# Pretraining Language Models with Text-Attributed Heterogeneous Graphs

**Tao Zou**[1,*], **Le Yu**[1,*], **Yifei Huang**[1], **LeiLei Sun**[1,†] and **Bowen Du**[1,2,3]

[1]SKLSDE Lab, Beihang University, Beijing, China
[2] Zhongguancun Laboratory, Beijing, China
[3] School of Transportation Science and Engineering, Beihang University, Beijing, China
{zoutao, yule, yifeihuang, leileisun, dubowen}@buaa.edu.cn

## Abstract

In many real-world scenarios (e.g., academic networks, social platforms), different types of entities are not only associated with texts but also connected by various relationships, which can be abstracted as Text-Attributed Heterogeneous Graphs (TAHGs). Current pretraining tasks for Language Models (LMs) primarily focus on separately learning the textual information of each entity and overlook the crucial aspect of capturing topological connections among entities in TAHGs. In this paper, we present a new pretraining framework for LMs that explicitly considers the topological and heterogeneous information in TAHGs. Firstly, we define a context graph as neighborhoods of a target node within specific orders and propose a topology-aware pretraining task to predict nodes involved in the context graph by jointly optimizing an LM and an auxiliary heterogeneous graph neural network. Secondly, based on the observation that some nodes are text-rich while others have little text, we devise a text augmentation strategy to enrich textless nodes with their neighbors' texts for handling the imbalance issue. We conduct link prediction and node classification tasks on three datasets from various domains. Experimental results demonstrate the superiority of our approach over existing methods and the rationality of each design. Our code is available at https://github.com/Hope-Rita/THLM.

## 1 Introduction

Pretrained Language Models (PLMs) (Devlin et al., 2019; Yang et al., 2019; Brown et al., 2020; Lan et al., 2020) that built upon the Transformer architecture (Vaswani et al., 2017) have been successfully applied in various downstream tasks such as automatic knowledge base construction (Bosselut et al., 2019) and machine translation (Herzig et al., 2020). Due to the design of pretraining tasks (e.g.,

---

*Equal Contribution
†Corresponding Author

masked language modeling (Devlin et al., 2019), next-token prediction (Radford et al., 2018), autoregressive blank infilling (Du et al., 2022)), PLMs can learn general contextual representations from texts in the large-scale unlabelled corpus.

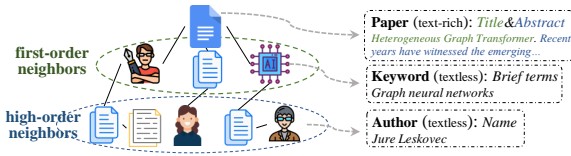

Figure 1: As an instance of TAHG, an academic network contains three types of nodes (papers, authors, and keywords) with textual descriptions as well as their multi-relational connections.

In fact, texts not only carry semantic information but also are correlated with each other, which could be well represented by Text-Attributed Heterogeneous Graphs (TAHGs) that include multi-typed nodes with textual descriptions as well as relations. See Figure 1 for an example. Generally, TAHGs usually exhibit the following two challenges that are struggled to be handled by existing PLMs.

*Abundant Topological Information (C1).* Both first- and higher-order connections exist in TAHGs and can reflect rich relationships. For instance, a paper can be linked to its references via first-order citations and can also be correlated with other papers through high-order co-authorships. However, the commonly used pretraining tasks (Radford et al., 2018; Devlin et al., 2019; Du et al., 2022) just learn from texts independently and thus ignore the connections among different texts. Although some recent works have attempted to make PLMs aware of graph topology (Yasunaga et al., 2022; Chien et al., 2022), they only consider first-order relationships and fail to handle higher-order signals.

*Imbalanced Textual Descriptions of Nodes (C2).* In TAHGs, nodes are heterogeneous and their carried texts are often in different magnitudes. For example, papers are described by both titles and

abstracts (rich-text nodes), while authors and keywords only have names or brief terms (textless nodes). Currently, how to pretrain LMs to comprehensively capture the above characteristics of TAHGs still remains an open question.

In this paper, we propose a new pretraining framework to integrate both **T**opological and **H**eterogeneous information in TAHGs into **LM**s, namely THLM. To address *C1*, we define a context graph as the neighborhoods of the central node within $K$ orders and design a topology-aware pretraining task (context graph prediction) to predict neighbors in the context graph. To be specific, we first obtain the contextual representation of the central node by feeding its texts into an LM and compute the structural representation of nodes in the given TAHG by an auxiliary heterogeneous graph neural network. Then, we predict which nodes are involved in the context graph based on the representations, aiming to inject the multi-order topology learning ability of graph neural networks into LMs. To tackle *C2*, we devise a text augmentation strategy, which enriches the semantics of textless nodes with their neighbors' texts and encodes the augmented texts by LMs. We conduct extensive experiments on three TAHGs from various domains to evaluate the model performance. Experimental results show that our approach could consistently outperform the state-of-the-art on both link prediction and node classification tasks. We also provide an in-depth analysis of the context graph prediction pretraining task and text augmentation strategy. Our key contributions include:

- We investigate the problem of pretraining LMs on a more complicated data structure, i.e., TAHGs. Unlike most PLMs that can only learn from the textual description of each node, we present a new pretraining framework to enable LMs to capture the topological connections among different nodes.

- We introduce a topology-aware pretraining task to predict nodes in the context graph of a target node. This task jointly optimizes an LM and an auxiliary heterogeneous graph neural network, enabling the LMs to leverage both first- and high-order signals.

- We devise a text augmentation strategy to enrich the semantics of textless nodes to mitigate the text-imbalanced problem.

## 2 Preliminaries

A **Pretrained Language Model (PLM)** can map an input sequence $X = (x_1, x_2, \cdots, x_L)$ of $L$ tokens into their contextual representations $\boldsymbol{H} = (\boldsymbol{h}_1, \boldsymbol{h}_2, \cdots, \boldsymbol{h}_L)$ with the design of pretraining tasks like masked language modeling (Devlin et al., 2019), next-token prediction (Radford et al., 2018), autoregressive blank infilling (Du et al., 2022). In this work, we mainly focus on the encoder-only PLMs (e.g., BERT (Devlin et al., 2019), RoBERTa (Liu et al., 2019)) and leave the explorations of PLMs based on encoder-decoder or decoder-only architecture in the future.

A **Text-Attributed Heterogeneous Graph (TAHG)** (Shi et al., 2019) usually consists of multi-typed nodes as well as different kinds of relations that connect the nodes. Each node is also associated with textual descriptions of varying lengths. Mathematically, a TAHG can be represented by $\mathcal{G} = (\mathcal{V}, \mathcal{E}, \mathcal{U}, \mathcal{R}, \mathcal{X})$, where $\mathcal{V}$, $\mathcal{E}$, $\mathcal{U}$ and $\mathcal{R}$ denote the set of nodes, edges, node types, and edge types, respectively. Each node $v \in \mathcal{V}$ belongs to type $\phi(v) \in \mathcal{U}$ and each edge $e_{u,v}$ has a type $\psi(e_{u,v}) \in \mathcal{R}$. $\mathcal{X}$ is the set of textual descriptions of nodes. Note that a TAHG should satisfy $|\mathcal{U}| + |\mathcal{R}| > 2$.

Existing PLMs mainly focus on textual descriptions of each node separately, and thus fail to capture the correlations among different nodes in TAHGs (as explained in Section 1). To address this issue, we propose a new framework for pretraining LMs with TAHGs, aiming to obtain PLMs that are aware of the graph topology as well as the heterogeneous information.

## 3 Methodology

Figure 2 shows the overall framework of our proposed approach, which mainly consists of two components: topology-aware pretraining task and text augmentation strategy. Given a TAHG, the first module extracts the context graph for a target node and predicts which nodes are involved in the context graph by jointly optimizing an LM and an auxiliary heterogeneous graph neural network. It aims to enable PLMs to capture both first-order and high-order topological information in TAHGs. Since some nodes may have little textual descriptions in TAHGs, the second component is further introduced to tackle the imbalanced textual descriptions of nodes, which enriches the semantics of textless nodes by neighbors' texts. It is worth notic-

ing that after the pretraining stage, we discard the auxiliary heterogeneous graph neural network and *only* apply the PLM for various downstream tasks.

## 3.1 Topology-aware Pretraining Task

To tackle the drawback that most existing PLMs cannot capture the connections between nodes with textual descriptions, some recent works have been proposed (Yasunaga et al., 2022; Chien et al., 2022). Although insightful, these methods solely focus on the modeling of first-order connections between nodes while ignoring high-order signals, which are proved to be essential in fields like network analysis (Grover and Leskovec, 2016; Cui et al., 2019), graph learning (Kipf and Welling, 2017; Hamilton et al., 2017) and recommender system (Wang et al., 2019; He et al., 2020). To this end, we propose a topology-aware pretraining task (namely, context graph prediction) for helping LMs capture multi-order connections among different nodes.

**Context Graph Extraction**. We first illustrate the definition of the context graph of a target node. Let $\mathcal{N}_u$ be the set of first-order neighbors of node $u$ in a given TAHG $\mathcal{G} = (\mathcal{V}, \mathcal{E}, \mathcal{U}, \mathcal{R}, \mathcal{X})$. The context graph $\mathcal{G}_u^K$ of node $u$ is composed of neighbors that $u$ can reach within $K$ orders (including node $u$ itself) as well as their connections, which is represented by $\mathcal{G}_u^K = (\mathcal{V}_u^K, \mathcal{E}_u^K)$. $\mathcal{V}_u^K = \{v'|v' \in \mathcal{N}_v \wedge v \in \mathcal{V}_u^{K-1}\} \cup \mathcal{V}_u^{K-1}$ is the node set of $\mathcal{G}_u^K$ and $\mathcal{E}_u^K = \{(u', v') \in \mathcal{E}|u' \in \mathcal{V}_u^K \wedge v' \in \mathcal{V}_u^K\}$) denotes the edge set of $\mathcal{G}_u^K$. It is obvious that $\mathcal{V}_u^0 = \{u\}$ and $\mathcal{V}_u^1 = \mathcal{N}_u \cup \{u\}$. Based on the definition, we can extract the context graph of node $u$ based on the given TAHG $\mathcal{G}$. Note that when $K \geq 2$, the context graph $\mathcal{G}_u^K$ will contain multi-order correlations between nodes, which provides an opportunity to capture such information by learning from $\mathcal{G}_u^K$.

**Context Graph Prediction**. TAHGs not only contain multiple types of nodes and relations but also involve textual descriptions of nodes. Instead of pretraining on single texts like most PLMs do, we present the Context Graph Prediction (CGP) for pretraining LMs on TAHGs to capture the rich information. Since LMs have been shown to be powerful in modeling texts (Devlin et al., 2019; Brown et al., 2020), the objective of CGP is to inject the graph learning ability of graph neural networks (Bing et al., 2022) into LMs.

Specifically, we first utilize an auxiliary heterogeneous graph neural network to encode the input

TAHG $\mathcal{G}$ and obtain the representations of all the nodes in $\mathcal{V}$ as follows,

$$\boldsymbol{H}^{\mathcal{G}} = f_{HGNN}(\mathcal{G}) \in \mathbb{R}^{|\mathcal{V}| \times d}, \qquad (1)$$

where $f_{HGNN}(\cdot)$ can be implemented by any existing heterogeneous graph neural networks. $d$ is the hidden dimension. Then, we encode the textual description of target node $u$ by an LM and derive its semantic representation by

$$\boldsymbol{h}_{LM}^u = \text{MEAN}(f_{LM}(X_u)) \in \mathbb{R}^d, \qquad (2)$$

where $f_{LM}(\cdot)$ can be realized by the existing LMs. Besides, to capture the heterogeneity of node $u$, we introduce a projection header in the last layer of the PLM. $X_u$ denotes the textual descriptions of node $u$. Next, we predict the probability that node $v$ is involved in the context graph $\mathcal{G}_u^K$ of $u$ via a binary classification task

$$\hat{y}_{u,v} = \text{sigmoid}\left(\boldsymbol{h}_{LM}^{u\top} \boldsymbol{W}_{\phi(v)} \boldsymbol{H}_v^{\mathcal{G}}\right), \qquad (3)$$

where $\boldsymbol{W}_{\phi(v)} \in \mathbb{R}^{d \times d}$ is a trainable transform matrix for node type $\phi(v) \in \mathcal{R}$. The ground truth $y_{u,v} = 1$ if $\mathcal{G}_u^K$ contains $v$, and 0 otherwise.

**Pretraining Process**. In this work, we use BERT (Devlin et al., 2019) and R-HGNN (Yu et al., 2022) to implement $f_{LM}(\cdot)$ and $f_{HGNN}(\cdot)$, respectively. Since it is intractable to predict the appearing probabilities of all the nodes $v \in \mathcal{V}$ in Equation (3), we adopt negative sampling (Mikolov et al., 2013) to jointly optimize $f_{LM}(\cdot)$ and $f_{HGNN}(\cdot)$. To generate positive samples, we uniformly sample $k$ neighbors from a specific relation during each hop. The negative ones are sampled from the remaining node set $\mathcal{V} \setminus \mathcal{V}_u^K$ with a negative sampling ratio of 5 (i.e., five negative samples per positive sample). In addition to the CGP task, we incorporate the widely used Masked Language Modeling (MLM) task to help LMs better handle texts. The final objective function for each node $u \in \mathcal{V}$ is

$$\mathcal{L}_u = \mathcal{L}_u^{MLM} + \mathcal{L}_u^{CGP} = -\log P(\tilde{X}_u|X_{u \setminus \tilde{X}_u}) -$$
$$\sum_{v \in \mathcal{V}_u^K} \log \hat{y}_{u,v} - \sum_{i=1}^5 \mathbb{E}_{v_i' \sim P_n(\mathcal{V} \setminus \mathcal{V}_u^K)} \log\left(1 - \hat{y}_{u,v_i'}\right), \qquad (4)$$

where $\tilde{X}_u$ is the corrupted version of node $u$'s original textual descriptions $X_u$ with a 40% masking rate following (Wettig et al., 2023). $P_n(\cdot)$ denotes the normal noise distribution. Additionally, the input feature of each node for the auxiliary heterogeneous graph neural network is initialized by

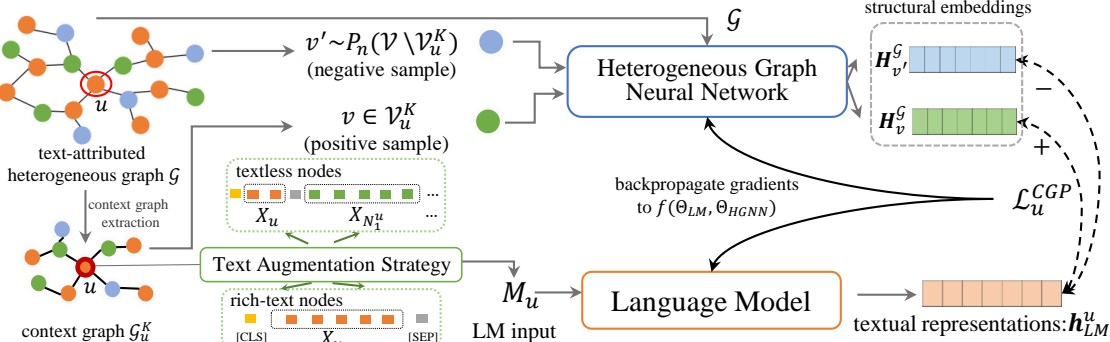

Figure 2: Framework of the proposed approach.

its semantic representation based on Equation (2) [1], which is shown to be better than a randomly-initialized trainable feature in the experiments.

## 3.2 Text Augmentation Strategy

As discussed in Section 1, the textual descriptions of different types of nodes in TAHGs are varying with different lengths, resulting in rich-text nodes and textless nodes. The exhaustive descriptions of rich-text nodes can well reveal their characteristics, while the brief descriptions of textless nodes are insufficient to reflect their semantics and solely encoding such descriptions would lead to suboptimal performance. Therefore, we devise a text augmentation strategy to tackle the imbalance issue, which first enriches the semantics of textless nodes by combining the textual descriptions of their neighbors according to the connections in TAHGs and then computes the augmented texts by LMs.

To be specific, for rich-text node $u$, we use its texts with special tokens (Devlin et al., 2019) as the input $M_u$, which is denoted as [CLS] $X_u$ [SEP]. For textless node $u$, we concatenate its texts and $k$ sampled neighbors' texts as the input $M_u$, i.e., [CLS] $X_u$ [SEP] $X_{\mathcal{N}_u^1}$ [SEP] ... [SEP] $X_{\mathcal{N}_u^k}$ [SEP],[2] where $\mathcal{N}_u^i$ represents the $i$-th sampled neighbor of $u$. Furthermore, in the case of nodes lacking text information, we employ the concatenation of text sequences from neighbors. This approach enables the generation of significant semantic representations for such nodes, effectively addressing the issue of text imbalance. After the augmentation of texts, we change the input of Equation (2) from $X_u$ to $M_u$ to obtain representation

$h_{LM}^u$ with more semantics. We empirically find that text augmentation strategy can bring nontrivial improvements without a significant increment of the model's complexity.

## 3.3 Fine-tuning in Downstream Tasks

After the pretraining process, we discard the auxiliary heterogeneous graph neural network $f_{HGNN}(\cdot)$ and *solely* apply the pretrained LM $f_{LM}(\cdot)$ to generate the semantic representations of nodes based on Equation (2). We select two graph-related downstream tasks for evaluation including link prediction and node classification. We employ various headers at the top of $f_{LM}(\cdot)$ to make exhaustive comparisons, including MultiLayer Perceptron (MLP), RGCN (Schlichtkrull et al., 2018), HetSANN (Hong et al., 2020), and R-HGNN (Yu et al., 2022). For downstream tasks, $f_{LM}(\cdot)$ is frozen for efficiency and only the headers can be fine-tuned. Please refer to the Appendix A.2 for detailed descriptions of the headers.

## 4 Experiments

### 4.1 Datasets and Baselines

**Datasets**. We conduct experiments on three real-world datasets from different domains, including the academic network (OAG-Venue (Hu et al., 2020b)), book publication (GoodReads (Wan and McAuley, 2018; Wan et al., 2019)), and patent application (Patents[3]). All the datasets have raw texts on all types of nodes, whose detailed descriptions and statistics are shown in the Appendix A.1.

**Compared Methods**. We compare THLM with several baselines to generate the representations of nodes and feed them into the headers for downstream tasks. In particular, we select six methods to

---

[1]Note that the initialization is executed *only once* by using the official checkpoint of BERT (Devlin et al., 2019).

[2]Among the neighbors in $\mathcal{N}_u$, we select rich-text nodes in priority. Moreover, if the size of $\mathcal{N}_u$ is smaller or equal to $k$, we will choose all the neighbors.

[3]https://www.uspto.gov/

compute the node representations: BERT (Devlin et al., 2019) and RoBERTa (Liu et al., 2019) are widely used PLMs; MetaPath (Dong et al., 2017) is a representative method for heterogeneous network embedding; MetaPath+BERT combines the textual and structural information as the representations, LinkBERT (Yasunaga et al., 2022) and GIANT (Chien et al., 2022) are first-order topology-aware PLMs. Besides, we apply OAG-BERT (Liu et al., 2022) to compare the performance of OAG-Venue. Detailed information about baselines is shown in the Appendix A.1. It is worth noticing that LinkBERT and GIANT are designed for homogeneous graphs instead of TAHGs. Hence, we maintain the 2-order connections among rich-text nodes and remove the textless nodes to build homogeneous graphs for these two methods for evaluation. See Appendix A.6 for more details.

## 4.2 Experimental Settings

Following the official configuration of $BERT_{base}$ (110M params, (Devlin et al., 2019)), we limit the input length of the text to 512 tokens. For the context graph prediction task, the number of orders $K$ in extracting context graphs is searched in $[1, 2, 3, 4]$. For the text augmentation strategy, we search the number of neighbors $k$ for concatenation in $[1, 2, 3]$. We load the weights in $BERT_{base}$ checkpoint released from Transformers tools[4] for initialization. For R-HGNN, we set the hidden dimension of node representations and relation representations to 786 and 64, respectively. The number of attention heads is 8. We use the two-layered R-HGNN in the experiments. To optimize THLM, we use AdamW (Loshchilov and Hutter, 2019) as the optimizer with $(\beta_1, \beta_2) = (0.9, 0.999)$, weight decay 0.01. For $BERT_{base}$, we warm up the learning rate for the first 8,000 steps up to 6e-5, then linear decay it. For R-HGNN, the learning rate is set to 1e-4. We set the dropout rate (Srivastava et al., 2014) of $BERT_{base}$ and R-HGNN to 0.1. We train for 80,000 steps, and batch sizes of 32, 48, and 64 sequences with 512 tokens for OAG-Venue, GoodReads, and Patents, and with maximize utilization while meeting the device constraints. The pretraining process took about three days on four GeForce RTX 3090 GPUs (24GB memory). For downstream tasks, please see Appendix A.4 for detailed settings of various headers.

---

[4] https://huggingface.co/bert-base-cased

## 4.3 Evaluation Tasks

**Link Prediction**. On OAG-Venue, GoodReads, and Patents, the predictions are between paper-author, book-publisher, and patent-company, respectively. We use RMSE and MAE as evaluation metrics, whose descriptions are shown in Appendix A.3. Considering the large number of edges on the datasets, we use a sampling strategy for link prediction. Specifically, the ratio of the edges used for training, validation, and testing is 30%, 10%, and 10% in all datasets. Each edge is associated with five/one/one negative edge(s) in the training/validation/testing stage.

**Node Classification**. We classify the category of papers, books, and patents in OAG-Venue, GoodReads, and Patents. We use Micro-Precision, Micro-Recall, Macro-Precision, Macro-Recall, and NDCG to evaluate the performance of different models. Descriptions of the five metrics are shown in Appendix A.3. Each paper in OAG-Venue only belongs to one venue, which could be formalized as a multi-class classification problem. Each patent or each book is categorized into one or more labels, resulting in multi-label classification problems.

## 4.4 Performance Comparison

Due to space limitations, we present the performance on RMSE and MAE for link prediction, as well as Micro-Precision and Micro-Recall for node classification, in Table 1. For the performance on Macro-Precision, Macro-Recall, and NDCG on three datasets in the node classification task, please refer to Appendix A.5. From Table 1 and Appendix A.5, we have the following conclusions.

Firstly, except for MetaPath, BERT and RoBERTa exhibit relatively poorer performance in link prediction across three datasets compared to other baselines. This suggests that incorporating the structural information from the graph can greatly enhance the performance of downstream link prediction tasks. Moreover, RoBERTa achieves notable performance in node classification when compared to other baselines. This implies that leveraging better linguistic representations can further improve the overall performance.

Secondly, we observe that MetaPath, which solely captures the network embeddings, performs the worst performance among the evaluated methods. However, when MetaPath is combined with semantic information, it achieves comparable or even superior performance compared to RoBERTa. This

Table 1: Performance of different methods on three datasets in two downstream tasks. The best and second-best performances are boldfaced and underlined. *: THLM significantly outperforms the best baseline with p-value < 0.05

| Datasets | Model | Link Prediction | | | | | | Node Classification | | | | | | | |
|---|---|---|---|---|---|---|---|---|---|---|---|---|---|---|---|
| | | RMSE | | | MAE | | | Micro-Precision(@1) | | | | Micro-Recall(@1) | | | |
| | | HetSANN | RGCN | R-HGNN | HetSANN | RGCN | R-HGNN | MLP | HetSANN | RGCN | R_HGNN | MLP | HetSANN | RGCN | R-HGNN |
| OAG-Veune | BERT | 0.1987 | 0.2149 | 0.1802 | 0.0648 | 0.0886 | 0.0447 | 0.2257 | 0.3146 | 0.3136 | 0.3473 | 0.2257 | 0.3146 | 0.3136 | 0.3473 |
| | RoBERTa | 0.1931 | 0.2152 | 0.1689 | 0.0635 | 0.0814 | 0.0400 | 0.2527 | 0.3193 | 0.3341 | 0.3516 | 0.2527 | 0.3193 | 0.3341 | 0.3516 |
| | MetaPath | 0.2199 | 0.2415 | 0.1946 | 0.0842 | 0.0972 | 0.0544 | 0.1132 | 0.2693 | 0.2851 | 0.3011 | 0.1132 | 0.2693 | 0.2851 | 0.3011 |
| | MetaPath+BERT | 0.2213 | 0.2149 | 0.1651 | 0.0981 | 0.0734 | 0.0377 | 0.2307 | 0.3311 | 0.3317 | 0.3472 | 0.2307 | 0.3311 | 0.3317 | 0.3472 |
| | LinkBERT* | 0.1867 | 0.2229 | 0.1739 | 0.0628 | 0.0892 | 0.0424 | 0.2278 | 0.3108 | 0.3115 | 0.3508 | 0.2278 | 0.3108 | 0.3115 | 0.3508 |
| | GIANT* | 0.2045 | 0.2022 | 0.1709 | 0.0730 | 0.0761 | 0.0408 | 0.2280 | 0.3116 | 0.3074 | 0.3274 | 0.2280 | 0.3116 | 0.3074 | 0.3274 |
| | OAG-BERT | 0.1918 | 0.2030* | 0.1772 | 0.0634 | 0.0744 | 0.0386 | 0.2577 | 0.3214 | 0.3152 | 0.3425 | 0.2577 | 0.3214 | 0.3152 | 0.3425 |
| | THLM | **0.1857*** | **0.1893*** | **0.1591*** | **0.0614*** | **0.0722*** | **0.0352*** | **0.2637*** | **0.3409*** | **0.3398*** | **0.3575*** | **0.2637*** | **0.3409*** | **0.3398*** | **0.3575*** |
| GoodReads | BERT | 0.1424 | 0.1738 | 0.1103 | 0.0408 | 0.0586 | 0.0190 | 0.7274 | 0.8238 | 0.8240 | 0.8396 | 0.6984 | 0.7909 | 0.7911 | 0.8061 |
| | RoBERTa | 0.1349 | 0.1268 | 0.1044 | 0.0360 | 0.0298 | 0.0189 | 0.7363 | 0.8271 | 0.8314 | 0.8404 | 0.7069 | 0.7941 | 0.7982 | 0.8069 |
| | MetaPath | 0.1782 | 0.1740 | 0.1520 | 0.0639 | 0.0639 | 0.0470 | 0.1492 | 0.6448 | 0.6479 | 0.6883 | 0.1432 | 0.6190 | 0.6220 | 0.6608 |
| | MetaPath+BERT | 0.1314 | 0.1195 | 0.1403 | 0.0325 | 0.0280 | 0.0300 | 0.7240 | 0.8258 | 0.8320 | 0.8396 | 0.6951 | 0.7928 | 0.7988 | 0.8061 |
| | LinkBERT* | 0.1471 | 0.1362 | 0.1135 | 0.0443 | 0.0396 | 0.0212 | 0.7131 | 0.8209 | 0.8259 | 0.8369 | 0.6846 | 0.7882 | 0.7930 | 0.8035 |
| | GIANT* | 0.1323 | 0.1179 | 0.1089 | 0.0375 | 0.0271 | 0.0271 | 0.7580 | 0.8250 | 0.8300 | 0.8391 | 0.7277 | 0.7921 | 0.7969 | 0.8057 |
| | THLM | **0.1206*** | **0.1159*** | **0.1000*** | **0.0286*** | **0.0271*** | **0.0162*** | **0.7769*** | **0.8399*** | **0.8437*** | **0.8496*** | **0.7459*** | **0.8102*** | **0.8134*** | **0.8157*** |
| Patents | BERT | 0.3274 | 0.3135 | 0.2764 | 0.1945 | 0.1829 | 0.1284 | 0.6248 | 0.6603 | 0.6910 | 0.6448 | 0.3791 | 0.4006 | 0.4192 | 0.3912 |
| | RoBERTa | 0.3149 | 0.2926 | 0.2585 | 0.1836 | 0.1545 | 0.1119 | 0.6380 | 0.6735 | 0.7022 | 0.6985 | 0.3871 | 0.4087 | 0.4261 | 0.4238 |
| | MetaPath | 0.4816 | 0.4842 | 0.4842 | 0.3372 | 0.3352 | 0.3353 | 0.1996 | 0.4385 | 0.4548 | 0.4654 | 0.1211 | 0.2660 | 0.2759 | 0.2824 |
| | MetaPath+BERT | 0.2922 | 0.2840 | 0.2371 | 0.1483 | 0.1440 | 0.0944 | 0.6243 | 0.6583 | 0.6881 | 0.6877 | 0.3788 | 0.3994 | 0.4175 | 0.4173 |
| | LinkBERT* | 0.3080 | 0.3033 | 0.2601 | 0.1803 | 0.1738 | 0.1142 | 0.6504 | 0.6749 | 0.7048 | 0.7075 | 0.3946 | 0.4095 | 0.4277 | 0.4293 |
| | GIANT* | 0.2734 | **0.2454** | 0.2276 | 0.1537 | 0.1238 | 0.0976 | 0.6508 | 0.6709 | 0.6992 | 0.6939 | 0.3949 | 0.4071 | 0.4242 | 0.4210 |
| | THLM | **0.2522*** | 0.2513 | **0.2190*** | **0.1233*** | **0.1210*** | **0.0848*** | **0.7066*** | **0.7159*** | **0.7324*** | **0.7363*** | **0.4287*** | **0.4344*** | **0.4444*** | **0.4467*** |

highlights the importance of incorporating both structural information and textual representations for each node to enhance overall performance.

Third, we note that LinkBERT and GIANT achieve superior results in the majority of metrics for link prediction. This highlights the advantage of learning textual representations that consider the graph structure. However, both GIANT and LinkBERT may not yield satisfactory results in node classification on the OAG-Venue and GoodReads. This could be attributed to two reasons: 1) these models primarily focus on first-order graph topology while overlooking the importance of high-order structures, which are crucial in these scenarios; 2) these models are designed specifically for homogeneous graphs and do not consider the presence of multiple types of relations within the graph. Consequently, their effectiveness is limited in TAHGs and may impede their performance.

Moreover, OAG-BERT demonstrates competitive results in link prediction and strong performance in node classification, thanks to its ability to capture heterogeneity and topology during pretraining. This can be attributed to its capability to learn the heterogeneity and topology of graphs. However, it should be noted that OAG-BERT primarily captures correlations between papers and their metadata, such as authors and institutions, overlooking high-order structural information. These findings highlight the importance of considering both graph structure and high-order relationships when developing models for graph-based tasks.

Finally, THLM significantly outperforms the existing models due to: 1) integrating multi-order graph topology proximity into language models, which enables the model to capture a more comprehensive understanding of the graph topology; 2) enhancing the semantic representations for textless nodes via aggregating the neighbors' textual descriptions, that generates more informative representations for textless nodes.

### 4.5 Analysis of Context Graph Prediction

To explore the impact of incorporating multi-order graph topology into language models, we conduct several experiments. These experiments aim to investigate the effects of both first- and high-order topology information, as well as the model's ability to capture structural information using R-HGNN. For the remaining experiments on the analysis of different components like CGP and the text augmentation strategy, we intentionally removed the MLM task to isolate its effects in THLM, namely THLM$^\star$ in Figure 3 and Table 2.

**Evaluation on Multi-order Topology Information**. To assess the significance of multi-order neighbors' topology, we vary the number of orders $K$ in extracting the context graph from 1 to 4. The corresponding results are illustrated in Figure 3. Besides, to examine the impact of high-order neighbors, we solely predict the 2-order neighbors in the context graph prediction task, as indicated by w/ 2-order CGP in Table 2.

From the results, it is evident that THLM

achieves superior performance when predicting multi-order neighbors compared to solely predicting 1-order or 2-order neighbors. This suggests that modeling both first- and high-order structures enables LMs to acquire more comprehensive graph topology. Additionally, we observe that THLM exhibits better results when $K$ is 2 in context graph prediction. However, its performance gradually declines as we predict neighbors in higher layers, potentially due to the reduced importance of topological information in those higher-order layers.

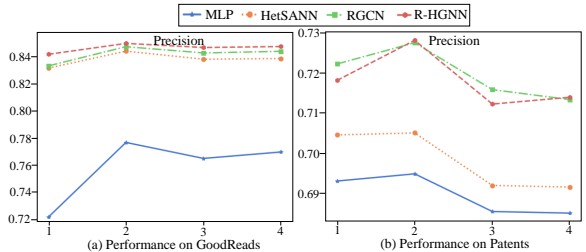

Figure 3: Effects of learning multi-order topology information in TAHGs on node classification.

Table 2: Evaluation of the ability to learn informative representations via R-HGNN on node classification

| Datasets | GCP | MLP | HetSANN | RGCN | R-HGNN |
|---|---|---|---|---|---|
| OAG-Venue | w/ MLP | 0.2591 | 0.3195 | 0.3043 | 0.3379 |
| | w/ RGCN | **0.2728** | 0.3323 | 0.3220 | 0.3547 |
| | w/ 2-order CGP | 0.2609 | 0.3357 | 0.3121 | 0.3488 |
| | w/ random feats | 0.2602 | 0.3271 | 0.3133 | 0.3487 |
| | THLM* | 0.2629 | **0.3383** | **0.3228** | **0.3554** |
| GoodReads | w/ MLP | 0.7528 | 0.8352 | 0.8376 | 0.8445 |
| | w/ RGCN | **0.7608** | 0.8380 | 0.8411 | **0.8512** |
| | w/ 2-order CGP | 0.7512 | 0.8319 | 0.8355 | 0.8431 |
| | w/ random feats | 0.7523 | **0.8384** | 0.8406 | 0.8483 |
| | THLM* | 0.7549 | 0.8382 | **0.8425** | 0.8485 |
| Patents | w/ MLP | 0.6903 | 0.6963 | 0.7201 | 0.7208 |
| | w/ RGCN | 0.6911 | 0.6986 | 0.7184 | 0.7218 |
| | w/ 2-order CGP | 0.6827 | 0.6876 | 0.7057 | 0.7068 |
| | w/ random feats | 0.6908 | 0.7001 | 0.7107 | 0.7198 |
| | THLM* | **0.6948** | **0.7050** | **0.7275** | **0.7280** |

**Evaluation on Learning Informative Node Features of R-HGNN**. In this work, we adopt one of the state-of-the-art HGNNs, i.e., R-HGNN with pre-initialized semantic features on nodes to obtain node representations. To examine the importance of learning informative node representations and complex graph structure in R-HGNN, we conduct experiments using two variants. Firstly, we replace R-HGNN with an MLP encoder or an alternative HGNN framework, i.e., RGCN (Schlichtkrull et al., 2018) in this experiment, denoted as w/ MLP and w/ RGCN respectively. Secondly, we substitute the semantic node features with randomly initialized trainable features, referred to as w/ random feats. The performance results are presented in Table 2.

From the obtained results, we deduce that both the initial features and effective HGNNs contribute significantly to capturing graph topology and embedding informative node representations effectively. Firstly, unlike MLP, which fails to capture the contextualized graph structure in the context graph prediction task, RGCN allows for the embedding of fine-grained graph structural information, which facilitates better learning of the graph topology. Furthermore, the utilization of effective HGNNs such as R-HGNN enables the embedding of expressive structural representations for nodes. Secondly, R-HGNN demonstrates its superior ability to learn more comprehensive graph structures from nodes compared to using randomly initialized features. These findings underscore the importance of integrating both semantic and structural information to learn informative node representations.

### 4.6 Analysis of Text Augmentation Strategy

Table 3: Results on the node classification task in evaluating the effectiveness of our text augmentation strategy.

| Dataset | Methods | MLP | HetSANN | RGCN | R-HGNN |
|---|---|---|---|---|---|
| OAG-Venue | neighbors-only | 0.2597 | 0.3274 | 0.3165 | 0.3495 |
| | textless-only | 0.2625 | 0.3290 | 0.3044 | 0.3516 |
| | TAS(1-Neighbor) | 0.2611 | 0.3349 | 0.3201 | 0.3507 |
| | TAS(2-Neighbor) | 0.2627 | 0.3380 | 0.3217 | 0.3549 |
| | TAS(3-Neighbor) | **0.2629** | **0.3383** | **0.3228** | **0.3554** |
| GoodReads | neighbors-only | 0.4855 | 0.7278 | 0.7132 | 0.7624 |
| | textless-only | 0.7453 | 0.8351 | 0.8397 | 0.8436 |
| | TAS(1-Neighbor) | 0.7480 | 0.8353 | 0.8421 | 0.8469 |
| | TAS(2-Neighbor) | 0.7547 | 0.8381 | **0.8426** | 0.8475 |
| | TAS(3-Neighbor) | **0.7549** | 0.8382 | 0.8425 | **0.8485** |
| Patents | neighbors-only | **0.6971** | 0.7040 | 0.7228 | 0.7224 |
| | textless-only | 0.6856 | 0.6923 | 0.7139 | 0.7164 |
| | TAS(1-Neighbor) | 0.6959 | 0.7004 | 0.7211 | 0.7221 |
| | TAS(2-Neighbor) | 0.6960 | **0.7050** | 0.7219 | 0.7233 |
| | TAS(3-Neighbor) | 0.6948 | **0.7050** | **0.7275** | **0.7281** |

To explore the potential of enhancing semantic information for textless nodes through our text augmentation strategy, we design three experimental variants. Firstly, we remove the text sequences of textless nodes and solely rely on the texts of their neighbors as inputs, denoted as "neighbors-only". We set the number of neighbors $k$ as 3 for concatenation. Secondly, we only use the original text descriptions of textless nodes to derive textual embeddings, namely "textless-only". Additionally, we employ the text augmentation strategy by varying the number of neighbors for concatenation from 1 to 3, denoted as "TAS(1-Neighbor)", "TAS(2-Neighbor)", and "TAS(3-Neighbor)", respectively. For all variants, we focus exclusively on the context graph prediction task to isolate the effects of

other factors. Due to space limitations, we present the Micro-Precision(@1) metric for node classification in the experiments. Similar trends could be observed across other metrics.

From Table 3, we observe that both neighbors and textless nodes themselves are capable of learning the semantic information for textless nodes. However, relying solely on either of them may lead to insufficient textual representations for nodes. Furthermore, it is found that using texts from more neighbors can enhance the semantic quality of textless nodes. Nevertheless, considering the limitations on the input sequence length of language models, we observe that THLM achieves similar performance when the number of $k$ is increased beyond 2. Therefore, to strike a balance between performance and computational efficiency while accommodating sequence length limitations, we choose $k$ as 3 for concatenation in the text augmentation strategy. To ensure the reliability of our findings, we conduct the task five times using different seeds ranging from 0 to 4. Remarkably, all obtained p-values are below 0.05, indicating statistical significance and confirming the accuracy improvement achieved by our text augmentation strategy.

### 4.7 Effects of Two Pretraining Tasks

To study the importance of two pretraining tasks for downstream tasks, we use two variants of THLM to conduct the experiments, and the performance is shown in Figure 4. Specifically, THLM w/o CGP removes the context graph prediction task, which does not predict the context neighbors for the input node. THLM w/o MLM reduces the masked language modeling task, which ignores the textual dependencies in the sentences and only predicts the multi-order graph topology in the pretraining process, i.e., by predicting the neighbors involved in the context graphs for input nodes.

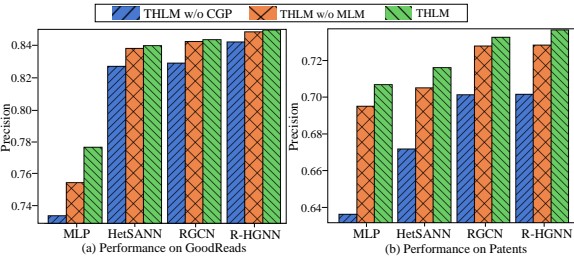

Figure 4: Importance of two pretraining tasks on the node classification task.

From Figure 4, we can conclude that THLM

achieves the best performance when it employs both two pretraining tasks for training. Removing either of these tasks leads to a decrease in the results. In particular, the context graph prediction task significantly contributes to the overall performance, demonstrating the substantial benefits of incorporating graph topology into our LM. Additionally, the masked language modeling task helps capture the semantics within texts better and further enhances the model performance. Besides, we find that THLM w/o MLM performs better than the original BERT on two datasets, which contributes to our text augmentation strategy for textless nodes. This enhancement allows for better connectivity between the brief terms of textless nodes and their neighboring text sequences, resulting in improved contextual understanding and representation in pretraining PLMs.

## 5 Related work

### 5.1 Pretrained Language Models

The objective of PLMs is to learn general representations of texts from large and unlabeled corpora via pretraining tasks, which could be applied to a variety of downstream tasks. Pretraining tasks that most PLMs widely used include 1) masked language modeling in BERT (Devlin et al., 2019) and RoBERTa (Liu et al., 2019); 2) next token prediction in GPT models (Radford et al., 2018; Brown et al., 2020); and 3) autoregressive blank infilling in GLM (Du et al., 2022). However, these tasks separately focus on the modeling within single texts and ignore the correlation among multiple texts.

Recently, several works have been proposed to capture the connections between different texts Levine et al. (2022); Chien et al. (2022); Yasunaga et al. (2022). For example, Chien et al. (2022) integrated the graph topology into LMs by predicting the connected neighbors of each node. Yasunaga et al. (2022) designed the document relation prediction task to pretrain LMs, which aims to classify the type of relation (contiguous, random, and linked) existing between two input text segments. Although insightful, these methods just consider the first-order connections between texts and cannot leverage high-order signals, which may lead to suboptimal performance. In this paper, we aim to present a new pretraining framework for LMs to help them comprehensively capture multi-order relationships as well as heterogeneous information in a more complicated data structure, i.e., TAHGs.

## 5.2 Heterogeneous Graph Learning

Graph Neural Networks (GNNs) (Kipf and Welling, 2017; Hamilton et al., 2017) have gained much progress in graph learning, which are extensively applied in modeling graph-structure data. Recently, many researchers have attempted to extend GNNs to heterogeneous graphs (Zhang et al., 2019; Fu et al., 2020; Hong et al., 2020; Yu et al., 2020; Hu et al., 2020b; Lv et al., 2021), which are powerful in handling different types of nodes and relations as well as the graph topological information. In this work, we aim to inject the graph learning ability of heterogeneous graph neural networks into PLMs via a topology-aware pretraining task.

## 5.3 Text-rich Network Mining

Many real-world scenarios (academic networks, patent graphs) can be represented by text-rich networks, where nodes are associated with rich text descriptions. Existing methods for text-rich network mining can be divided into two categories. The first branch designs the cascade architecture to learn the textual information by Transformer (Vaswani et al., 2017) and network topology by graph neural networks separately (Zhu et al., 2021; Li et al., 2021; Pang et al., 2022). Another group nests GNNs into LMs to collaboratively explore the textual and topological information (Yang et al., 2021; Jin et al., 2022, 2023a,b). However, these works either mainly focus on the homogeneous graph or modify the architecture of LMs by incorporating extra components. For example, Heterformers (Jin et al., 2023b) is developed for text-rich heterogeneous networks, which aims to embed nodes with rich text and their one-hop neighbors by leveraging the power of both LMs and GNNs during pretraining and downstream tasks. Different from these works, we learn about the more complicated TAHGs and employ auxiliary heterogeneous graph neural networks to assist LMs in capturing the rich information in TAHGs. After the pretraining, we discard the auxiliary networks and only apply the pretrained LMs for downstream tasks without changing their original architectures.

## 6 Conclusion

In this paper, we pretrained language models on more complicated text-attributed heterogeneous graphs, instead of plain texts. We proposed the context graph prediction task to inject the graph learning ability of graph neural networks into LMs, which jointly optimizes an auxiliary graph neural network and an LM to predict which nodes are involved in the context graph. To handle imbalanced textual descriptions of different nodes, a text augmentation strategy was introduced, which enriches the semantics of textless nodes by combining their neighbors' texts. Experimental results on three datasets showed that our approach could significantly and consistently outperform existing methods across two downstream tasks.

## 7 Limitations

In this work, we pretrained language models on TAHGs and evaluated the model performance on link prediction and node classification tasks. Although our approach yielded substantial improvements over baselines, there are still several promising directions for further investigation. Firstly, we just focused on pretraining encoder-only LMs, and it is necessary to validate whether encoder-decoder or decoder-only LMs can also benefit from the proposed pretraining task. Secondly, more downstream tasks that are related to texts (e.g., retrieval and reranking) can be compared in the experiments. Thirdly, it is interesting to explore the pretraining of LMs in larger scales on TAHGs.

## 8 Acknowledgements

This work was supported by the National Natural Science Foundation of China (51991395, 62272023), and the Fundamental Research Funds for the Central Universities (No. YWF-23-L-717, No. YWF-23-L-1203).

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

# A Appendix

## A.1 Datasets and Baselines

**Datasets.** Specific statistics of datasets are shown in Table 4 and detailed descriptions of datasets are shown as follows.

- **OAG-Venue**: OAG-Venue[5] is a heterogeneous graph followed by Hu et al. (2020b), which includes papers (P), authors (A), fields (F) and institutions (I). Each paper is published in a single venue. We treat papers as rich-text nodes and extract the title and abstract parts as their text descriptions. Authors, fields, and institutions are regarded as textless nodes, whose text descriptions are composed of their definitions or names.

- **GoodReads**: Following (Wan and McAuley, 2018; Wan et al., 2019), we receive a subset of GoodReads[6], which contains books (B), authors (A) and publishers (P). Each book is categorized into one or more genres. We treat books as rich-text nodes and extract brief introductions as their text descriptions. Authors and publishers are regarded as textless nodes, whose text descriptions are their names.

- **Patents**: Patents is a heterogeneous graph collected from the USPTO[7], which contains patent documents (P), applicants (A) and applied companies (C). Each patent is assigned several International Patent Classification (IPC) codes. We treat patents as rich-text nodes and extract the title and abstract parts as their text descriptions. Applicants and companies use their names as text descriptions, regarded as textless nodes.

**Baselines.** We compare our model with the following baselines: BERT (Devlin et al., 2019) and RoBERTa (Liu et al., 2019) are popular encoder-only pretraining language models. MetaPath (Dong et al., 2017) leverages meta-path-based random walks in the heterogeneous graph to generate node embeddings. MetaPath+BERT combines the textual embeddings embedded from BERT$_{base}$ and structural representations learned from MetaPath as node features. LinkBERT (Yasunaga et al., 2022) captures the dependencies across documents by predicting the relation between two segments on Wikipedia and BookCorpus. GIANT (Chien et al., 2022) extracts graph-aware node embeddings from raw text data via neighborhood prediction in the graph. OAG-BERT (Liu et al., 2022) is a pretrained language model specialized in academic knowledge services, allowing for the incorporation of heterogeneous entities such as authors, institutions, and keywords into paper embeddings.

## A.2 Headers in Downstream Tasks

We apply four methods on downstream tasks, which could be shown as follows,

- **MLP** relies exclusively on node features as input and uses the multilayer perceptron for prediction, which does not consider the graph information.

- **RGCN** incorporates the different relationships among nodes by using transformation matrices respectively in the knowledge graphs (Schlichtkrull et al., 2018).

- **HetSANN** aggregates different types of relations information from neighbors with a type-aware attention mechanism (Hong et al., 2020).

- **R-HGNN** learns the relation-aware node representation by integrating fine-grained representation on each set of nodes within separate relations, and semantic representations across different relations (Yu et al., 2022).

## A.3 Evaluation Metrics

Seven metrics are adopted to comprehensively evaluate the performance of different models in link prediction and node classification. In link prediction, we use Root Mean Square Error (RMSE) and Mean Absolute Error (MAE) metrics. In node classification, we use Micro-Precision, Micro-Recall, Macro-Precision, Macro-Recall, and Normalized Discounted Cumulative Gain (NDCG) metrics for evaluation. Details of the metrics are shown below.

RMSE evaluates the predicted ability for truth values, which calculates the error between prediction results and truth values. Given the prediction for all examples $\hat{y} = \{\hat{y}_1, \hat{y}_2, \cdots, \hat{y}_m\}$, and the truth data $y = \{y_1, y_2, \cdots, y_m\}$, we calculate the

---

[5]https://github.com/UCLA-DM/pyHGT
[6]https://sites.google.com/eng.ucsd.edu/ucsdbookgraph/home
[7]https://www.uspto.gov/

Table 4: Statistics of the datasets.

| Datasets | Nodes | Edges | Average Text Length | Category | Classification Split Sets | Link Prediction Split Sets |
|---|---|---|---|---|---|---|
| OAG-Venue | # Paper (P): 167,004
# Author (A): 511,122
# Field (F): 45,775
# Institution (I): 9,090 | # P-F: 1,709,601
# P-P: 864,019
# A-I: 614,161
# P-A: 480,104 | P: 243.497
A: 5.667
F: 3.690
I: 5.882 | 242 | Train: 106,724
Validation: 24,433
Test: 35,847 | Train: 144,030
Validation: 48,010
Test: 48,010 |
| GoodReads | # Book (B): 364,115
# Author (A): 154,418
# Publisher (P): 40,135 | # B-A: 572,654
# B-P: 466,626 | B: 163.577
A: 4.100
P: 5.120 | 8 | Train: 254,880
Validation: 54,617
Test: 54,618 | Train: 139,988
Validation: 46,662
Test: 46,662 |
| Patents | # Patent (P): 363,528
# Applicant (A): 182,561
# Company (C): 1,000 | # P-C: 367,598
# P-A: 334,906 | P: 139.436
A: 6.418
C: 8.436 | 565 | Train: 254,469
Validation: 54,529
Test: 54,530 | Train: 110,277
Validation: 36,759
Test: 36,759 |

total RMSE as follows,

$$\text{RMSE}(\hat{y}, y) = \sqrt{\frac{1}{m} \sum_{i=1}^{m} (\hat{y}_i - y_i)^2}.$$

MAE measures the absolute errors between predictions and truth values. Given the prediction for all examples $\hat{y} = \{\hat{y}_1, \hat{y}_2, \cdots, \hat{y}_m\}$, and the truth data $y = \{y_1, y_2, \cdots, y_m\}$, we calculate the total MAE as follows,

$$\text{MAE}(\hat{y}, y) = \frac{1}{m} \sum_{i=1}^{m} |\hat{y}_i - y_i|.$$

Micro-averaged precision measures the ability that recognizes more relevant elements than irrelevant ones in all classes. We select the top-K predicted labels as predictions for each sample. Hence, Micro-Precision@K is the proportion of positive predictions that are correct over all classes, which is calculated by,

$$\text{Micro-Precision@K} = \frac{\sum_{c_i \in C} \text{TP}(c_i)}{\sum_{c_i \in C} \text{TP}(c_i) + \text{FP}(c_i)},$$

where $\text{TP}(c_i)$, $\text{FP}(c_i)$ is the number of true positives, and false positives for class $c_i$ respectively.

Micro-averaged recall evaluates the model's ability in selecting all the relevant elements in all classes. We select the top-K probability predicted labels as predictions for each sample. Hence, Micro-Recall@K is the proportion of positive labels that are correctly predicted over all classes, which is calculated by,

$$\text{Micro-Recall@K} = \frac{\sum_{c_i \in C} \text{TP}(c_i)}{\sum_{c_i \in C} \text{TP}(c_i) + \text{FN}(c_i)},$$

where $\text{TP}(c_i)$, $\text{FN}(c_i)$ is the number of true positives, and false negatives for class $c_i$ respectively.

Macro-averaged precision reflects the average ability to recognize the relevant elements rather than irrelevant ones in each class. We select the top-K probability predicted labels as predictions. Hence, Macro-Precision@K is calculated by averaging all the precision values of all classes,

$$\text{Macro-Precision@K} = \frac{\sum_{c_i \in C} \text{P}(\hat{S}, S, c_i)}{|C|},$$

where $\hat{S}$, $S$ represents the predicted values and truth labels in the datasets, $\text{P}(\hat{S}, S, c_i)$ is the precision value of class $c_i$.

Macro-averaged recall evaluates the average ability to select all the relevant elements in each class. We select the top-K probability predicted labels as predictions. Hence, Macro-Recall@K is calculated by averaging all the recall values of all classes,

$$\text{Macro-Recall@K} = \frac{\sum_{c_i \in C} \text{R}(\hat{S}, S, c_i)}{|C|},$$

where $\hat{S}$, $S$ represents the predicted values and truth labels in the datasets, $\text{R}(\hat{S}, S, c_i)$ is the recall value of class $c_i$.

NDCG measures the ranking quality by considering the orders of all labels. For each sample $p_i$, NDCG is calculated by

$$\text{NDCG@K}(p_i) = \frac{\sum_{k=1}^{K} \frac{\delta(\hat{S}_i^k, S_i)}{\log_2(k+1)}}{\sum_{k=1}^{\min(K, |S_i|)} \frac{1}{\log_2(k+1)}},$$

where $\hat{S}_i^k$ denotes the $k$-th predicted label of example $p_i$. $\delta(v, S)$ is 1 when element $v$ is in set $S$, otherwise 0. We calculate the average NDCG of all examples as a metric.

## A.4 Detailed Settings in Downstream Tasks

In downstream tasks, we search the hidden dimension of node representation for headers in $[32, 64, 128, 256, 512]$. For methods that use attention mechanisms, (i.e., HetSANN and R-HGNN), the number of attention heads is searched in $[1, 2, 4, 8, 16]$. The training process is following R-HGNN (Yu et al., 2022).

## A.5 Detailed Experimental Results

We show the Macro-Precision(@1) and Macro-Recall(@1) in the node classification task on three datasets in Table 7. Since the values of NDCG(@1) are the same as Micro-Precision(@1), we do not show duplicate results. Besides, since node classification tasks on GoodReads and Patents belong to multi-label node classification, we show the performance on five metrics when K is 3 and 5 in Table 8 and Table 9 respectively.

## A.6 LinkBERT & GIANT

In our baselines, LinkBERT and GIANT are specifically designed for homogeneous text-attributed graphs, which cannot be directly applied in TAHGs. To address this, we convert the TAHGs into homogeneous graphs that contain the set of rich-text nodes and their connections to ensure that all nodes contain rich semantic information in the graphs. For Patents and GoodReads, we extract the 2-order relationships in the graph and discard the textless nodes along with their relative edges to construct the homogeneous graphs. In the case of the OAG-Venue dataset, due to the high density of the second-order graph, we choose to construct a homogeneous graph using a subset of crucial meta-path information to save the graph topology as much as possible. Inspired by Yu et al. (2022). we utilize the meta-path "P-F-P" (Paper-Field-Paper) and the direct relation "P-P" (Paper-Paper) to build the homogeneous graph for conducting experiments.

In addition to previous experiments, we conducted another experiment to capture the first-order information in the TAHGs while preserving the graph topology as much as possible. Specifically, we discard the heterogeneity of nodes and relationships in the graph to build a homogeneous graph, and the results are shown in Table 5.

From Table 5, it is evident that pretraining LinkBERT and GIANT on TAHGs solely for 1-order prediction may not yield optimal results. There are two key reasons for this observation: 1)

Table 5: Performance on node classification in LinkBERT and GIANT.

| Datasets | Model | Micro-Precision(@1) | | | |
|---|---|---|---|---|---|
| | | MLP | HetSANN | RGCN | R-HGNN |
| GoodReads | LinkBERT(1-order) | 0.6790 | 0.8100 | 0.8044 | 0.8302 |
| | GIANT(1-order) | 0.6967 | 0.8247 | 0.8284 | 0.8398 |
| | THLM | **0.7769** | **0.8399** | **0.8437** | **0.8496** |
| Patents | LinkBERT(1-order) | 0.5972 | 0.6421 | 0.6773 | 0.6734 |
| | GIANT(1-order) | 0.4793 | 0.6234 | 0.6323 | 0.6391 |
| | THLM | **0.7066** | **0.7159** | **0.7324** | **0.7363** |

textless nodes always lack sufficient textual content, leading to scarce semantic information. Hence, predicting relationships between textless nodes and their neighbors becomes challenging for language models. 2) Apart from first-order neighbors, high-order neighbors provide more complex structure information within the graph. By considering the relationships beyond the immediate neighbors, LMs could capture the graph topology across nodes more effectively and comprehensively. These findings highlight the importance of considering both first-order and high-order structure information in TAHGs and addressing the challenges of limited semantics on textless nodes. By tackling both problems, our model can learn better in TAHGs.

## A.7 Effect of Distinguishing Treasured Structural Information

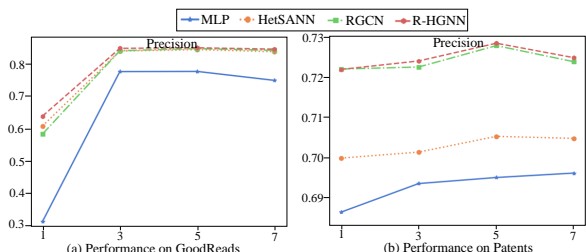

Figure 5: Precision on node classification with different numbers in sampling negative candidates in the pretraining process.

We investigate the effect of treasured structural information in the TAHGs. Specifically, we solely change the number of negative candidates for each positive entity in the context graph prediction task in $[1, 3, 5, 7]$ in the pretraining stage. We present the performance of GoodReads and Patents on the Micro-Precision(@1) metric in the node classification task.

From Figure 5, we could observe that the performance with a smaller number or larger number in sampling negative candidates would be worse. This observation can be explained by two factors.

Firstly, the model may receive limited structural information when selecting a smaller number of negative candidates, which hampers the model's ability to understand the underlying topology structure effectively. Secondly, sampling a larger number of negative candidates may bring noise topological information and make it difficult to distinguish meaningful patterns and relationships. Hence, the optimal performance is achieved when the number of sampled negative candidates falls within a proper range. By striking a balance between learning sufficient topological information and avoiding excessive noise, the model can effectively capture the graph structure and achieve better performance in downstream tasks.

### A.8 Performance on Large-scale Datasets

In our evaluation, we further test THLM on large-scale datasets (i.e., obgn-mag dataset (Hu et al., 2020a)) for the node classification task. The performance is shown in Table 6. We observe that THLM demonstrates scalability to larger datasets, outperforming baselines such as LinkBERT and GIANT. This outcome highlights the effectiveness of THLM, particularly its superior performance on the obgn-mag dataset.

Table 6: The accuracy results for node classification on the obgn-mag dataset.

| Model | MLP | HetSANN | RGCN |
|---|---|---|---|
| BERT | 0.3754 | 0.5298 | 0.5484 |
| RoBERTa | 0.3770 | 0.5300 | 0.5490 |
| LinkBERT* | 0.3775 | 0.5230 | 0.5491 |
| GIANT* | 0.3903 | 0.5184 | 0.5256 |
| THLM | **0.3933** | **0.5353** | **0.5517** |

Table 7: Performance of different methods on three datasets in node classification. The best and second-best performances are boldfaced and underlined.

| Datasets | Model | Macro-Precision(@1) ↑ | | | | Macro-Recall(@1) ↑ | | | |
|---|---|---|---|---|---|---|---|---|---|
| | | MLP | HetSANN | RGCN | R-HGNN | MLP | HetSANN | RGCN | R-HGNN |
| OAG-Venue | BERT | 0.2110 | 0.3104 | 0.3119 | 0.3359 | 0.1992 | 0.3118 | 0.3060 | 0.3415 |
| | RoBERTa | 0.2429 | 0.3264 | 0.3258 | **0.3598** | 0.2387 | 0.3187 | 0.3169 | 0.3412 |
| | MetaPath | 0.0959 | 0.2593 | 0.2830 | 0.3005 | 0.0717 | 0.2731 | 0.2663 | 0.3019 |
| | MetaPath+BERT | 0.2094 | 0.3202 | 0.3248 | 0.3363 | 0.1991 | 0.3150 | 0.3180 | 0.3368 |
| | LinkBERT⋆ | 0.2054 | 0.2921 | 0.3014 | 0.3479 | 0.2060 | 0.3057 | 0.2902 | 0.3233 |
| | GIANT⋆ | 0.2026 | 0.3078 | 0.3080 | 0.3381 | 0.2005 | 0.3097 | 0.2858 | 0.3188 |
| | THLM | **0.2506** | **0.3375** | **0.3408** | 0.3562 | **0.2464** | **0.3330** | **0.3331** | **0.3537** |
| GoodReads | BERT | 0.7352 | 0.8273 | 0.8253 | 0.8421 | 0.7040 | 0.7960 | 0.7969 | 0.8112 |
| | RoBERTa | 0.7420 | 0.8290 | 0.8328 | 0.8428 | 0.7134 | 0.7994 | 0.8039 | 0.8120 |
| | MetaPath | 0.1786 | 0.6599 | 0.6560 | 0.6966 | 0.1371 | 0.6204 | 0.6225 | 0.6624 |
| | MetaPath+BERT | 0.7285 | 0.8286 | 0.8356 | 0.8425 | 0.7015 | 0.7978 | 0.8026 | 0.8104 |
| | LinkBERT⋆ | 0.7178 | 0.8239 | 0.8276 | 0.8389 | 0.6917 | 0.7932 | 0.7987 | 0.8091 |
| | GIANT⋆ | 0.7622 | 0.8273 | 0.8329 | 0.8418 | 0.7331 | 0.7970 | 0.8018 | 0.8109 |
| | THLM | **0.7798** | **0.8472** | **0.8493** | **0.8515** | **0.7516** | **0.8148** | **0.8184** | **0.8209** |
| Patents | BERT | 0.3526 | 0.3876 | 0.4073 | 0.2994 | 0.1587 | 0.1864 | 0.1795 | 0.1335 |
| | RoBERTa | 0.3262 | 0.3918 | 0.4185 | 0.4227 | 0.1506 | 0.1941 | 0.1801 | 0.1846 |
| | MetaPath | 0.0854 | 0.1894 | 0.1862 | 0.2059 | 0.0153 | 0.0941 | 0.0930 | 0.0946 |
| | MetaPath+BERT | 0.3330 | 0.3827 | 0.4072 | 0.4263 | 0.1577 | 0.1866 | 0.1842 | 0.1929 |
| | LinkBERT⋆ | 0.3458 | 0.3838 | 0.4182 | 0.4515 | 0.1649 | 0.1858 | 0.1884 | 0.1920 |
| | GIANT⋆ | 0.3506 | 0.3904 | 0.4194 | 0.4327 | 0.1764 | 0.1995 | 0.1928 | 0.1944 |
| | THLM | **0.4374** | **0.4364** | **0.4466** | **0.4974** | **0.2090** | **0.2128** | **0.2115** | **0.2281** |

Table 8: Performance of different methods on GoodReads in node classification. The best and second-best performances are boldfaced and underlined.

| Metric | Model | K=3 | | | | K=5 | | | |
|---|---|---|---|---|---|---|---|---|---|
| | | MLP | HetSANN | RGCN | R-HGNN | MLP | HetSANN | RGCN | R-HGNN |
| Macro-Precision | BERT | 0.3402 | 0.3645 | 0.3520 | 0.3637 | 0.2146 | 0.2193 | 0.2095 | 0.2136 |
| | RoBERTa | 0.3418 | 0.3602 | 0.3552 | 0.3707 | 0.2145 | 0.2174 | 0.2104 | 0.2166 |
| | MetaPath | 0.1463 | 0.3374 | 0.3283 | 0.3417 | 0.1415 | 0.2187 | 0.2107 | 0.2142 |
| | MetaPath+BERT | 0.3377 | 0.3676 | 0.3605 | **0.3749** | 0.2138 | 0.2202 | 0.2137 | 0.2182 |
| | LinkBERT* | 0.3350 | 0.3688 | 0.3543 | 0.3647 | 0.2118 | 0.2209 | 0.2114 | 0.2133 |
| | GIANT* | **0.3526** | 0.3671 | 0.3609 | 0.3702 | **0.2186** | 0.2187 | **0.2146** | **0.2189** |
| | THLM | 0.3458 | **0.3753** | **0.3647** | 0.3717 | 0.2139 | **0.2232** | 0.2125 | 0.2136 |
| Macro-Recall | BERT | 0.9368 | 0.9766 | 0.9755 | 0.9804 | 0.9814 | 0.9941 | 0.9935 | 0.9947 |
| | RoBERTa | 0.9431 | 0.9791 | 0.9792 | 0.9819 | 0.9851 | 0.9948 | 0.9945 | 0.9952 |
| | MetaPath | 0.3950 | 0.8608 | 0.8585 | 0.8863 | 0.6461 | 0.9445 | 0.9395 | 0.9554 |
| | MetaPath+BERT | 0.9357 | 0.9762 | 0.9788 | 0.9803 | 0.9806 | 0.9939 | 0.9949 | 0.9950 |
| | LinkBERT* | 0.9283 | 0.9756 | 0.9760 | 0.9785 | 0.9768 | 0.9938 | 0.9935 | 0.9942 |
| | GIANT* | 0.9502 | 0.9766 | 0.9775 | 0.9803 | 0.9862 | 0.9947 | 0.9945 | 0.9951 |
| | THLM | **0.9615** | **0.9829** | **0.9846** | **0.9836** | **0.9899** | **0.9961** | **0.9959** | **0.9957** |
| Micro-Precision | BERT | 0.3252 | 0.3391 | 0.3386 | 0.3403 | 0.2046 | 0.2071 | 0.2070 | 0.2072 |
| | RoBERTa | 0.3274 | 0.3399 | 0.3399 | 0.3409 | 0.2053 | 0.2072 | 0.2072 | 0.2073 |
| | MetaPath | 0.1438 | 0.2999 | 0.2991 | 0.3086 | 0.1410 | 0.1976 | 0.1964 | 0.1995 |
| | MetaPath+BERT | 0.3249 | 0.3389 | 0.3399 | 0.3404 | 0.2044 | 0.2071 | 0.2073 | 0.2073 |
| | LinkBERT* | 0.3222 | 0.3388 | 0.3388 | 0.3397 | 0.2037 | 0.2071 | 0.2070 | 0.2071 |
| | GIANT* | 0.3299 | 0.3390 | 0.3393 | 0.3404 | 0.2056 | 0.2072 | 0.2072 | 0.2073 |
| | THLM | **0.3338** | **0.3413** | **0.3418** | **0.3414** | **0.2063** | **0.2075** | **0.2075** | **0.2074** |
| Micro-Recall | BERT | 0.9368 | 0.9767 | 0.9753 | 0.9802 | 0.9821 | 0.9942 | 0.9936 | 0.9947 |
| | RoBERTa | 0.9430 | 0.9790 | 0.9791 | 0.9820 | 0.9854 | 0.9948 | 0.9945 | 0.9954 |
| | MetaPath | 0.4142 | 0.8638 | 0.8614 | 0.8888 | 0.6767 | 0.9484 | 0.9427 | 0.9578 |
| | MetaPath+BERT | 0.9357 | 0.9762 | 0.9791 | 0.9805 | 0.9813 | 0.9941 | 0.9952 | 0.9952 |
| | LinkBERT* | 0.9281 | 0.9758 | 0.9758 | 0.9785 | 0.9777 | 0.9941 | 0.9936 | 0.9943 |
| | GIANT* | 0.9502 | 0.9764 | 0.9774 | 0.9804 | 0.9868 | 0.9948 | 0.9946 | 0.9953 |
| | THLM | **0.9613** | **0.9831** | **0.9846** | **0.9835** | **0.9902** | **0.9962** | **0.9960** | **0.9957** |
| NDCG | BERT | 0.8526 | 0.9164 | 0.9158 | 0.9252 | 0.8713 | 0.9236 | 0.9233 | 0.9312 |
| | RoBERTa | 0.8600 | 0.9192 | 0.9209 | 0.9266 | 0.8776 | 0.9257 | 0.9273 | 0.9321 |
| | MetaPath | 0.3008 | 0.7740 | 0.7735 | 0.8070 | 0.4089 | 0.8088 | 0.8072 | 0.8354 |
| | MetaPath+BERT | 0.8507 | 0.9170 | 0.9214 | 0.9253 | 0.8695 | 0.9244 | 0.9280 | 0.9314 |
| | LinkBERT* | 0.8413 | 0.9149 | 0.9168 | 0.9231 | 0.8617 | 0.9224 | 0.9241 | 0.9295 |
| | GIANT* | 0.8732 | 0.9168 | 0.9195 | 0.9252 | 0.8883 | 0.9243 | 0.9266 | 0.9313 |
| | THLM | **0.8879** | **0.9288** | **0.9310** | **0.9314** | **0.8998** | **0.9342** | **0.9357** | **0.9364** |

Table 9: Performance of different methods on Patents in node classification. The best and second-best performances are boldfaced and underlined.

| Metric | Model | K=3 | | | | K=5 | | | |
|---|---|---|---|---|---|---|---|---|---|
| | | MLP | HetSANN | RGCN | R-HGNN | MLP | HetSANN | RGCN | R-HGNN |
| Macro-Precision | BERT | 0.2012 | 0.2502 | 0.2634 | 0.2365 | 0.1425 | 0.1800 | 0.1883 | 0.1866 |
| | RoBERTa | 0.2010 | 0.2421 | 0.2648 | 0.2886 | 0.1414 | 0.1725 | 0.1836 | 0.2136 |
| | MetaPath | 0.0655 | 0.1321 | 0.1389 | 0.1579 | 0.0523 | 0.1062 | 0.1102 | 0.1234 |
| | MetaPath+BERT | 0.2041 | 0.2541 | 0.2626 | 0.2914 | 0.1418 | 0.1818 | 0.1860 | 0.2098 |
| | LinkBERT* | 0.2144 | 0.2443 | 0.2715 | 0.2933 | 0.1490 | 0.1758 | 0.1877 | 0.2194 |
| | GIANT* | 0.2181 | 0.2459 | 0.2692 | 0.2854 | 0.1518 | 0.1799 | 0.1882 | 0.2137 |
| | THLM | **0.2541** | **0.2671** | **0.2827** | **0.3133** | **0.1761** | **0.1864** | **0.1950** | **0.2300** |
| Macro-Recall | BERT | 0.3036 | 0.3553 | 0.3592 | 0.2765 | 0.3824 | 0.4326 | 0.4493 | 0.3619 |
| | RoBERTa | 0.3017 | 0.3598 | 0.3603 | 0.3618 | 0.3827 | 0.4430 | 0.4560 | 0.4526 |
| | MetaPath | 0.0335 | 0.1889 | 0.1949 | 0.1884 | 0.0484 | 0.2446 | 0.2543 | 0.2465 |
| | MetaPath+BERT | 0.3027 | 0.3603 | 0.3610 | 0.3682 | 0.3810 | 0.4383 | 0.4522 | 0.4527 |
| | LinkBERT* | 0.3186 | 0.3597 | 0.3714 | 0.3699 | 0.4038 | 0.4483 | 0.4631 | 0.4568 |
| | GIANT* | 0.3344 | 0.3591 | 0.3713 | 0.3654 | 0.4131 | 0.4324 | 0.4616 | 0.4511 |
| | THLM | **0.3933** | **0.4023** | **0.4067** | **0.4111** | **0.4890** | **0.4886** | **0.5038** | **0.4976** |
| Micro-Precision | BERT | 0.3502 | 0.3636 | 0.3785 | 0.3599 | 0.2426 | 0.2502 | 0.2590 | 0.2488 |
| | RoBERTa | 0.3566 | 0.3694 | 0.3845 | 0.3826 | 0.2472 | 0.2541 | 0.2626 | 0.2615 |
| | MetaPath | 0.1286 | 0.2580 | 0.2695 | 0.2729 | 0.0971 | 0.1874 | 0.1941 | 0.1962 |
| | MetaPath+BERT | 0.3498 | 0.3646 | 0.3773 | 0.3775 | 0.2428 | 0.2507 | 0.2582 | 0.2584 |
| | LinkBERT* | 0.3609 | 0.3699 | 0.3841 | 0.3851 | 0.2494 | 0.2542 | 0.2625 | 0.2627 |
| | GIANT* | 0.3596 | 0.3656 | 0.3804 | 0.3775 | 0.2484 | 0.2505 | 0.2600 | 0.2583 |
| | THLM | **0.3843** | **0.3863** | **0.3951** | **0.3959** | **0.2626** | **0.2627** | **0.2684** | **0.2686** |
| Micro-Recall | BERT | 0.6375 | 0.6618 | 0.6890 | 0.6552 | 0.7360 | 0.7591 | 0.7856 | 0.7548 |
| | RoBERTa | 0.6491 | 0.6724 | 0.6998 | 0.6964 | 0.7499 | 0.7709 | 0.7968 | 0.7933 |
| | MetaPath | 0.2341 | 0.4697 | 0.4905 | 0.4967 | 0.2945 | 0.5684 | 0.5888 | 0.5951 |
| | MetaPath+BERT | 0.6367 | 0.6636 | 0.6868 | 0.6871 | 0.7367 | 0.7606 | 0.7834 | 0.7838 |
| | LinkBERT* | 0.6570 | 0.6734 | 0.6992 | 0.7010 | 0.7567 | 0.7711 | 0.7963 | 0.7970 |
| | GIANT* | 0.6546 | 0.6655 | 0.6923 | 0.6871 | 0.7537 | 0.7598 | 0.7888 | 0.7835 |
| | THLM | **0.6996** | **0.7032** | **0.7192** | **0.7207** | **0.7967** | **0.7970** | **0.8144** | **0.8148** |
| NDCG | BERT | 0.6725 | 0.7025 | 0.7297 | 0.6921 | 0.7066 | 0.7353 | 0.7610 | 0.7262 |
| | RoBERTa | 0.6854 | 0.7140 | 0.7417 | 0.7387 | 0.7200 | 0.7470 | 0.7726 | 0.7697 |
| | MetaPath | 0.2467 | 0.4902 | 0.5103 | 0.5188 | 0.2734 | 0.5296 | 0.5487 | 0.5573 |
| | MetaPath+BERT | 0.6717 | 0.7024 | 0.7272 | 0.7280 | 0.7066 | 0.7351 | 0.7586 | 0.7594 |
| | LinkBERT* | 0.6953 | 0.7154 | 0.7413 | 0.7444 | 0.7291 | 0.7478 | 0.7725 | 0.7748 |
| | GIANT* | 0.6936 | 0.7084 | 0.7354 | 0.7305 | 0.7275 | 0.7397 | 0.7665 | 0.7617 |
| | THLM | **0.7442** | **0.7488** | **0.7652** | **0.7675** | **0.7752** | **0.7785** | **0.7950** | **0.7963** |