# OpenReview forum: "Pretraining Language Models with Text-Attributed Heterogeneous Graphs"
_EMNLP/2023/Conference — EMNLP 2023 Findings_

### Official Review · Reviewer_v8Uk · 2023-08-04

**Soundness:** 3

**Excitement:**

3: Ambivalent: It has merits (e.g., it reports state-of-the-art results, the idea is nice), but there are key weaknesses (e.g., it describes incremental work), and it can significantly benefit from another round of revision. However, I won't object to accepting it if my co-reviewers champion it.

**Missing References:**

Zhao J, Qu M, Li C, et al. Learning on large-scale text-attributed graphs via variational inference[J]. arXiv preprint arXiv:2210.14709, 2022.

**Paper Topic And Main Contributions:**

This paper proposes a new pre-training framework for language models for Text-Attributed Heterogeneous Graphs (TAHGs). This method considers the topological relationships between texts in the heterogeneous graph during the pre-training of the language model, and introduces a text augmentation strategy to enhance the semantics of textless nodes.



**Questions For The Authors:**

A. In Section 2.1, the process of sampling k neighbors is not detailed. Could you clarify how this sampling is conducted?
B. In Table 1, the authors reconstruct homogeneous graphs for GIANT and LinkBert. In terms of link prediction tasks, how is a fair comparison maintained with the original heterogeneous graphs?
C. There are inconsistencies in the experimental section that need to be clarified by the authors. For instance, the results of THLM in Tables 2 and 3 differ from those in Table 1. THLM without CGP approximates BERT, but there's a significant discrepancy between the blue column for the Patents data in Figure 4 and the results for BERT in Table 1. Could the authors please explain this?


**Reasons To Accept:**

1. Text-Attributed Graph Learning is a significant area of study. Both the textual modality and the graph structure have unique characteristics, and pre-training a language model (LM) using a Heterogeneous Graph Neural Network (HGNN) is a challenging task.
2. For predicting the nodes in the context graph of a target node, a topology-aware pre-training task is designed.
3. The method leverages higher-order information from heterogeneous graphs, a feature that is relatively overlooked in homogeneous graphs. This point is corroborated in the experiments through comparison with homogeneous graph methods, such as GIANT and LinkBERT.
4. The presentation is well-structured and easy to follow.


**Reasons To Reject:**

1. Beyond Table 1, results on the OAG-Venue dataset are noticeably absent from the experimental analysis in the main text of the paper.
2. The technical novelty of the paper is somewhat restricted, as both Masked Language Modeling (MLM) and Heterogeneous graph neural network (HGNN) are widely studied and extensively documented in existing literature.
3. In Equation 3, the authors introduced a learnable matrix W_(ϕ(v)) to handle the heterogeneity of node v. However, it is unclear how the heterogeneity of node u is accounted for.
4. The authors have only evaluated mid-sized datasets in this paper. It would be beneficial for the authors to consider including results from experiments on the ogbn-mag dataset to bolster the persuasiveness of their findings.


**Reproducibility:**

4: Could mostly reproduce the results, but there may be some variation because of sample variance or minor variations in their interpretation of the protocol or method.

**Reviewer Confidence:**

4: Quite sure. I tried to check the important points carefully. It's unlikely, though conceivable, that I missed something that should affect my ratings.

---

> ### Author Rebuttal · Authors · 2023-08-29
>
> Thank you for the valuable feedback. We appreciate your attention to the performance of our proposed approach on the OAG-Venue and MAG datasets and we show the performance in the following tables. Additionally, we have highlighted the novelty of our work and provided a thorough explanation of our design. Furthermore, we further clarified the performance results in the experimental section. These additions aim to enhance the overall clarity and comprehensibility of our paper.
>
> **W1: Experiments on OAG-Venue.**
>
> We appreciate your feedback on the performance of our approach on the OAG-Venue dataset. We acknowledge that due to space limitations, we did not provide the performance on OAG-Venue in the remaining experiments, except for Table 1. However, we would like to inform you that we have included the performance (Micro-Precision on node classification) about the evaluation of Context Graph Prediction (Section 4.5) and ablation study (Section 4.7) in the following table, and similar observations can be made for other metrics as well. These results and the performance of the rest of the experiments align with the original paper's findings. We will ensure that all of the results on the OAG-Venue dataset are included in future revisions or updates of the paper.
>
>
> | Methods         | MLP        | HetSANN    | RGCN       | RHGNN      |
> | --------------- | ---------- | ---------- | ---------- | ---------- |
> | w/ MLP          | 0.2591     | 0.3195     | 0.3043     | 0.3379     |
> | w/ RGCN         | 0.2728     | 0.3323     | 0.322      | 0.3547     |
> | w/ 2-order CGP  | 0.2609     | 0.3357     | 0.3121     | 0.3488     |
> | w/ random feats |    0.2602        |   0.3271         |    0.3133        |     0.3487       |
> | THLM w/o MLM   | **0.2629** | **0.3383** | **0.3228** | **0.3554** |
>
>
> | Methods      | MLP    | HetSANN | RGCN   | RHGNN  |
> | ------------ | ------ | ------- | ------ | ------ |
> | THLM         | **0.2637** | **0.3409**  | **0.3398** | **0.3575** |
> | THLM w/o CGP | 0.2260 | 0.3275  | 0.3152 | 0.3485 |
> | THLM w/o MLM | 0.2629 | 0.3383  | 0.3228 | 0.3554 |
>
> **W2: Explanation of the novelty of THLM.**
>
> We propose a novel pretraining framework for Language Models (LMs) that incorporates topological and heterogeneous information from Text-Attributed Heterogeneous Graphs (TAHGs). Unlike existing approaches such as LinkBERT and GIANT, which focus on capturing first-order connections between texts, our method emphasizes learning both first-hop and high-order topology in TAHGs. We also introduce a text augmentation strategy to enhance the representations of textless nodes by aggregating neighbors' text information.
>
> In contrast to GLEM, where LMs and Graph Neural Networks (GNNs) are jointly used for latent representation inference, our approach utilizes Heterogeneous Graph Neural Networks (HGNNs) as auxiliary modules to incorporate graph topology into PLMs. Our goal is to integrate the graph learning capability of HGNNs into PLMs through a topology-aware pretraining task. After the pretraining phase, we discard the auxiliary networks and solely rely on the pretrained LMs, without modifying their original architectures. This allows us to effectively utilize the learned representations in various downstream tasks.
>
> Overall, we aim to present a new pretraining framework for LMs to help them comprehensively capture multi-order relationships as well as heterogeneous information in a more complicated data structure, i.e., TAHGs
>
> **W3: Design for the heterogeneity of node u in Equation 3.**
>
> In our approach, to capture the heterogeneity of node $u$, we introduce a projection header in the last layer of the PLM, which is utilized in Equation 2. The process involves multiplying $h^u_{LM}$ with $W_{\phi(u)}$, similar to how we handle the heterogeneity of node $v$. We acknowledge that the embedding process in Equation 3 could be clarified further, and we will ensure to provide a more detailed explanation in future versions of the paper.
>
> **W4: Performance on ogbn-mag datasets.**
>
> In our evaluation, we tested THLM on the obgn-mag dataset for the node classification task. Considering the limited time available, we utilized hyperparameters from the OAG-Venue dataset, and we acknowledge that further hyperparameter searches are desirable for future experiments. The results of THLM on the test set, specifically the Micro-Precision on node classification, are presented in the following table.
>
> We observed that THLM demonstrates scalability to larger datasets, outperforming baselines such as LinkBERT and GIANT. This outcome highlights the effectiveness of THLM, particularly its superior performance on the obgn-mag dataset.
>
>
> | Model    | MLP    | HetSANN | RGCN   |
> | -------- | ------ | ------- | ------ |
> | BERT     | 0.3754 | 0.5298  | 0.5484 |
> | RoBERTa  | 0.3770 | 0.5300  | 0.5490 |
> | LinkBERT | 0.3775 | 0.5230  | 0.5491 |
> | GIANT    | 0.3903 | 0.5184  | 0.5256 |
> | THLM     | **0.3933** | **0.5353**  | **0.5517** |
>
> **Q1: Explanation of the sampling process.**
>
> After carefully reviewing our work, we find that there is no explicit mention of the sampling process in Section 2.1. To provide a clear understanding of the sampling process, we would like to provide detailed explanations of both sampling processes employed in our work.
>
> For our text augmentation strategy, we prioritize the utilization of text sampled from rich-text neighbors. We do this by uniformly selecting text from these neighbors, thereby enriching the textual information associated with textless nodes. However, in cases where there are no rich-text neighbors available around textless nodes, we resort to concatenating the text from neighboring textless nodes as an alternative strategy.
>
> Regarding the context graph prediction task, we adopt a uniform sampling strategy. Specifically, during each hop, we uniformly sample $k$ neighbors from a specific relation. These selected neighbors from all kinds of relations are considered positive candidates during the training process. In order to generate negative candidates, we employ a negative sampling strategy.
>
> **Q2: Comparison between THLM and LinkBERT, GIANT.**
>
> Regarding the LinkBERT and GIANT models, we followed a specific procedure for their pretraining and downstream tasks. We pretrained these models on reconstructed homogeneous graphs, generating contextual representations for all nodes based on their respective text sequences. However, during downstream tasks, we utilized the original heterogeneous graphs for prediction.
>
> Due to the nature of LinkBERT and GIANT, which are designed for homogeneous networks with rich-text nodes, we encountered a challenge on datasets such as Patents and GoodReads. In these datasets, there are no direct connections between rich-text nodes. To address this, we constructed a homogeneous graph by selecting connections up to 2 hops among rich-text nodes. This approach aimed to preserve the original topology as much as possible. These models are denoted as LinkBERT(\*) and GIANT(\*). However, it's important to note that the reconstructed graph overlooks the presence of textless nodes and may result in some loss of graph topology. To further investigate the impact of retaining the original topology, we conducted another experiment in which we discarded the heterogeneity of nodes and relationships in the original heterogeneous graph to build a homogeneous graph for pre-training, namely LinkBERT(1-hop) and GIANT(1-hop). The performance on link prediction and node classification is shown respectively as follows (we use RMSE and Micro-Precision metric on these two tasks, other metrics have similar observations).
>
>
> | Dataset   | Model           | HetSANN | RGCN   | R-HGNN |
> | --------- | --------------- | ------- | ------ | ------ |
> | GoodReads | LinkBERT(1-hop) | 0.1507  | 0.1423 | 0.1189 |
> |           | LinkBERT(*)     | 0.1471  | 0.1362 | 0.1135 |
> |           | GIANT(1-hop)    | 0.1365  | 0.1259 | 0.1066 |
> |           | GIANT(*)        | 0.1323  | 0.1179 | 0.1089 |
> |           | THLM            | **0.1206**  | **0.1159** | **0.1000** |
> | Patents   | LinkBERT(1-hop) | 0.3122  | 0.3079 | 0.2641 |
> |           | LinkBert(*)     | 0.3080  | 0.3033 | 0.2601 |
> |           | GIANT(1-hop)    | 0.2781  | 0.2492 | 0.2309 |
> |           | GIANT(*)        | 0.2734  | **0.2454** | 0.2276 |
> |           | THLM            | **0.2522**  | 0.2513 | **0.2190** |
>
> | Dataset   | Model           | MLP    | HetSANN | RGCN   | R-HGNN |
> | --------- | --------------- | ------ | ------- | ------ | ------ |
> | GoodReads | LinkBERT(1-hop) | 0.6790 | 0.8100  | 0.8044 | 0.8302 |
> |           | LinkBERT(*)     | 0.7244 | 0.8209  | 0.8259 | 0.8369 |
> |           | GIANT(1-hop)    | 0.6967 | 0.8247  | 0.8284 | 0.8398 |
> |           | GIANT(*)        | 0.6887 | 0.8098  | 0.8150 | 0.8258 |
> |           | THLM            | **0.7769** | **0.8439**  | **0.8437** | **0.8496** |
> | Patents   | LinkBERT(1-hop) | 0.5972 | 0.6421  | 0.6773 | 0.6734 |
> |           | LinkBERT(*)     | 0.6333 | 0.6670  | 0.6968 | 0.6957 |
> |           | GIANT(1-hop)    | 0.4793 | 0.6234  | 0.6323 | 0.6391 |
> |           | GIANT(*)        | 0.6508 | 0.6701  | 0.6956 | 0.6999 |
> |           | THLM            | **0.7066** | **0.7159**  | **0.7324** | **0.7363** |
>
> From the results, it is evident that pretraining LinkBERT and GIANT on TAHGs solely for 1-order prediction may not yield optimal results. There are two key reasons for this observation: 1) The homogeneous graph used in LinkBERT and GIANT is built on rich-text nodes. They focus on capturing the relationships among rich-text nodes, which are the second-hop relationships in our graph. Hence, they fail to learn the graph topology well with the original heterogeneous graph. 2) Textless nodes always lack sufficient textual content, leading to scarce semantic information. Hence, predicting relationships between textless nodes and their neighbors becomes challenging for language models.  These findings highlight the importance of considering both first-order and high-order structure information in TAHGs and addressing the challenges of limited semantics on textless nodes. By tackling both problems, our model can learn better in TAHGs. (See Appendix A.6 for detailed settings of GIANT and LinkBERT experiments)
>
>
> **Q3 (Part 1/2): Clarification on the performance in Table 1, Table 2, and Table 3.**
>
> In Table 1 presented in our original paper, alongside the CGP task, we introduced the MLM task to improve the LM's ability to capture semantic information on text sequences. However, for the remaining experiments, involving the analysis of different components like CGP and the text augmentation strategy, we intentionally removed the MLM task to isolate its effects. Therefore, the performance results of THLM in the subsequent experiments, such as in Table 2 and Table 3 were obtained without the MLM task, as mentioned in Lines 451-452 of the paper. We will provide further elaboration in the revised version of the paper.
>
>
> **Q3 (Part 2/2): Clarification on the performance between THLM and BERT.**
>
> The distinction between THLM without MLM and the original BERT stems from our text augmentation strategy for textless nodes. In the original BERT, we simply masked the brief terms of textless nodes. However, in THLM without MLM, we take a different approach by enhancing the semantics of textless nodes. This enhancement allows for better connectivity between the brief terms of textless nodes and their neighboring text sequences, resulting in improved contextual understanding and representation in pre-training PLMs.

---

### Official Review · Reviewer_xFg8 · 2023-08-05

**Soundness:** 4

**Excitement:**

3: Ambivalent: It has merits (e.g., it reports state-of-the-art results, the idea is nice), but there are key weaknesses (e.g., it describes incremental work), and it can significantly benefit from another round of revision. However, I won't object to accepting it if my co-reviewers champion it.

**Missing References:**

OAG-BERT [1] mentioned above

**Paper Topic And Main Contributions:**

This paper proposes a language model pre-training framework for text-attributed heterogeneous graphs (TAHGs). The key idea is to leverage not only text semantics but also topological connections between texts during pre-training. The authors propose to tackle two challenges, higher-order connections and imbalanced text information, which are less concerned in previous studies. To be specific, they train an auxiliary heterogeneous graph neural network and devise a text augmentation strategy to solve the two challenges, respectively. Experiments on three datasets and two tasks demonstrate the effectiveness of the proposed THLM model.

**Questions For The Authors:**

- Could you repeat your experiments (the fine-tuning stage) multiple times, conduct a t-test, and report the p-values by comparing THLM with the strongest baseline in each column?

- Could you compare THLM with OAG-BERT (on OAG-Venue only) and GraphFormers?

- Can THLM deal with nodes without any text information?

**Reasons To Accept:**

+ The idea of exploiting graph information in heterogeneous networks has practical value due to the prevalence of structural information in real-world texts.

+ Different from previous studies (e.g., LinkBERT and GraphFormers) that considers first-order and homogeneous links only, this paper identifies the challenges of tackling higher-order and heterogeneous structures. The proposed framework of combining pre-trained language models and heterogeneous graph neural networks is intuitive and well-motivated.

+ The authors conduct comprehensive ablation analyses and hyperparameter studies to validate their design choices and configurations.

**Reasons To Reject:**

- The novelty of this work is not significant. Modeling higher-order network information has been extensively explored in (heterogeneous) GNN studies. Considering imbalanced text information in heterogeneous networks has been studied in Heterformer. There are also many related works combining language models and graph structures together using different architectures. Therefore, this work is more like integrating many existing techniques together.

- Significance tests are missing. It is unclear whether the improvement of THLM over baselines and ablation versions is statistically significant or not. In fact, some gaps between THLM and baselines/ablation versions are quite subtle in Tables 1-3, therefore p-values should be reported.

- Some relevant baselines, such as GraphFormers and OAG-BERT [1], are not compared in the experiments.

[1] OAG-BERT: Towards a Unified Backbone Language Model for Academic Knowledge Services. KDD 2022.

**Reproducibility:**

4: Could mostly reproduce the results, but there may be some variation because of sample variance or minor variations in their interpretation of the protocol or method.

**Reviewer Confidence:**

4: Quite sure. I tried to check the important points carefully. It's unlikely, though conceivable, that I missed something that should affect my ratings.

---

> ### Author Rebuttal · Authors · 2023-08-29
>
> Thank you for your comments. We have strengthened the novelty of our work, and have conducted extensive experiments comparing THLM with baselines. Additionally, we have run OAG-BERT and done a comparison with THLM. At last, we elaborated on the process of dealing with nodes with no text in our model. These efforts will undoubtedly contribute to the overall strength and significance of my work.
>
> **W1: Novelty of our work.**
>
> In this paper, we present a novel pretraining framework for Language Models (LMs) that explicitly incorporates the topological and heterogeneous information found in Text-Attributed Heterogeneous Graphs (TAHGs). We compare our model with existing works from two perspectives:
>
> * Unlike traditional Heterogeneous Graph Neural Networks (HGNNs) that focus on learning node representations, our objective is to integrate the graph learning capabilities of HGNNs into PLMs through a topology-aware pretraining task. To achieve this, we employ representative HGNNs to capture the graph topology, enabling LMs to leverage both first- and high-order graph structures.
>
> * Existing approaches for modeling text-rich networks typically concentrate on homogenous networks or modify the architecture of LMs by incorporating external components. In our work, we employ an auxiliary HGNN during LM pretraining, but subsequently discard the auxiliary networks and solely utilize the pretrained LMs for downstream tasks without altering their original architectures.
>
> It should be noted that Heterformers are specifically designed for text-attributed networks, with a focus on embedding rich-text nodes and their surrounding neighbors. However, they explore the graph structure by combining LMs and GNNs in both pretraining and downstream tasks. Additionally, Heterformers may fail to capture the graph topology around textless nodes, limiting their ability to effectively learn the graph structure. To address this challenge and mitigate the text-imbalanced problem, we employ a text augmentation strategy that enriches the semantics of textless nodes. This augmentation strategy facilitates the learning of the graph structure around these nodes, enhancing the overall performance of our approach.
>
>
>
> **W2&Q1: Significance tests on THLM and baselines.**
>
> We conducted significance tests to compare the performance of THLM with the strongest baseline in both link prediction and node classification tasks. In order to present the results within the given space limitations, we specifically showcase the p-values for the MAE and Micro-Precision metrics associated with these two tasks. Notably, all the calculated p-values are found to be less than 0.05, indicating significant superiority of our THLM model over the baselines. These results validate the effectiveness and improved performance of our approach in both link prediction and node classification tasks.
>
> | p-values  | HetSANN   | RGCN      | RHGNN     |
> | --------- | --------- | --------- | --------- |
> | Books     | 3.161E-05 | 2.907E-04 | 1.629E-05 |
> | Patents   | 2.059E-06 | 1.862E-05 | 4.131E-05 |
> | OAG-Venue | 4.512E-04 | 3.024E-04 | 1.783E-05 |
>
> | p-values  | MLP       | HetSANN   | RGCN      | RHGNN     |
> | --------- | --------- | --------- | --------- | --------- |
> | Books     | 5.985E-05 | 1.461E-06 | 1.107E-04 | 2.669E-07 |
> | Patents   | 5.220E-07 | 1.460E-06 | 1.070E-06 | 3.270E-05 |
> | OAG-Venue | 1.476E-05 | 2.212E-05 | 2.124E-03 | 5.983E-05 |
>
> It is important to note that the performance evaluation of THLM in Table 1 and Table 2-3 is subtly nuanced. This is because we report the performance of THLM without MLM (Masked Language Modeling) in all remaining experiments, except for the results presented in Table 1 to isolate its effects.  Consequently, the performance of THLM in Table 2 and Table 3 actually refers to THLM without MLM. We acknowledge the need for clarification on this matter, and we will provide a more detailed explanation in the revised version of the paper.
>
>
> **W3&Q2: Comparison with OAG-BERT and GraphFormer.**
>
> Thank you for providing insights into the related works. OAG-BERT is a pre-trained language model specialized in academic knowledge services, allowing for the incorporation of heterogeneous entities such as authors, institutions, and keywords into paper embeddings. On the other hand, GraphFormers introduces a GNN-nest Transformer framework to simultaneously capture the graph structure surrounding different papers and the semantics of their texts.
>
> We show the performance of OAG-BERT and other models in the following table. However, it is worth noting that in our work, we encountered challenges when attempting to use GraphFormers. The requirement of pretraining the PLM with text information from the center node and its surrounding neighbors demands significant computational resources. As a result, we were unable to successfully run GraphFormers within the given time constraints. We plan to conduct experiments using GraphFormers in the future and include the results in our revised paper.
>
> |     Model     | HetSANN |  RGCN  | R-HGNN |
> | :-----------: | :-----: | :----: | :----: |
> |     BERT      | 0.1987  | 0.2149 | 0.1802 |
> |    Roberta    | 0.1931  | 0.2152 | 0.1689 |
> |   MetaPath    | 0.2199  | 0.2415 | 0.1946 |
> | MetaPath+BERT | 0.2213  | 0.2149 | 0.1651 |
> |   LinkBERT    | 0.1867  | 0.2229 | 0.1739 |
> |     GIANT     | 0.2045  | 0.2022 | 0.1709 |
> |   OAG-BERT    | 0.1918  | 0.2030 | 0.1772 |
> |     THLM      | **0.1857**  | **0.1893** | **0.1591** |
>
> | Model         | MLP    | HetSANN | RGCN   | R-HGNN |
> | ------------- | ------ | ------- | ------ | ------ |
> | BERT          | 0.2242 | 0.3146  | 0.3136 | 0.3473 |
> | Roberta       | 0.2527 | 0.3193  | 0.3341 | 0.3516 |
> | MetaPath      | 0.1132 | 0.2693  | 0.2851 | 0.3011 |
> | MetaPath+BERT | 0.2307 | 0.3311  | 0.3317 | 0.3472 |
> | LinkBERT      | 0.2278 | 0.3108  | 0.3110 | 0.3508 |
> | GIANT         | 0.2280 | 0.3116  | 0.3074 | 0.3274 |
> | OAG-BERT      | 0.2577 | 0.3214  | 0.3152 | 0.3425 |
> | THLM          | **0.2637** | **0.3409**  | **0.3398** | **0.3575** |
>
> Upon analyzing the results, it is evident that OAG-BERT achieves competitive results in link prediction when compared to LinkBERT and GIANT and demonstrates strong performance compared to the baselines in node classification. This can be attributed to its ability to learn the heterogeneity and topology of graphs during the pretraining phase. However, it is important to note that OAG-BERT primarily captures correlations between papers and their meta-data information, such as authors, venues, and institutions, while disregarding the high-order structural information. As a result, OAG-BERT performs relatively worse than our model, which takes into account the high-order structure and achieves improved performance in the task at hand.
>
> **Q3: Dealing with nodes without text sequences.**
>
> In our experimental setup, we encountered nodes that did not have any text information associated with them. To address this issue, we implemented the text augmentation strategy by concatenating the text sequences from neighboring nodes. This approach allows us to generate meaningful semantic representations for these nodes, effectively mitigating the text-imbalanced problem. By incorporating the text information from neighboring nodes, THLM can handle nodes without any text descriptions in a consistent manner.

---

### Official Review · Reviewer_m9df · 2023-08-06

**Soundness:** 3

**Excitement:**

3: Ambivalent: It has merits (e.g., it reports state-of-the-art results, the idea is nice), but there are key weaknesses (e.g., it describes incremental work), and it can significantly benefit from another round of revision. However, I won't object to accepting it if my co-reviewers champion it.

**Paper Topic And Main Contributions:**

This paper proposes a pre-training framework for heterogeneous graphs where each node is associated with texts. The existing LMs can learn the textual information well but they ignore the graph topology which could be important to perform well the representation learning for nodes and to eventually achieve high performance on downstream tasks. The proposed framework, THLM, basically attempts to address two problems: 1) how to learn graph structure with high-order topological information and 2) how to tackle imbalanced textual information of nodes. For the first one, it extracts a context graphs for each node and the a GNN model is trained so as to predict its neighbor nodes involved in the context graph. To enrich the textual information for a node with lack of text, the framework simply concatenates its neighbor nodes' texts and use them as input to LM model. The authors conducted extensive experiments, including three different datasets and six baseline methods, to demonstrate the effectiveness of the framework. For both node classification and link prediction which are used as downstream tasks, THLM achieves higher accuracy overall compared to the baselines. Also, the paper shows the impact of the text augmentation strategy and the impact of different orders in context graph extraction.

**Questions For The Authors:**

* What's the main idea to deal with heterogeneous node types effectively?
* What is A in line 144?
* Is there something that makes it challenging to apply THLM  to encoder-decoder and decoder only models?

**Reasons To Accept:**

* The framework is able to incorporate both graph structural information and textual information into learning node representations even when the node types are heterogeneous.
* The framework seems to be easily applicable to various foundation GNN/LM models.
* The experiments were conducted extensively. They cover various aspects of the proposed framework.

**Reasons To Reject:**

* It is not quite convincing how much effective and efficient the text augmentation strategy is. Compared to the case where only textless node's own texts are used, the accuracy gain is not significant. Also, the increase in the amount of texts by taking neighbors' texts would make the overall computation efficiency worse.
* Not sure if the graph topology is really well learned. Given a node, the framework draws samples without any consideration about how the selected neighbors are connected to the node. It just considers whether the sampled neighbor is part of K-order neighbors or not.
* The paper claims the proposed framework is for heterogeneous graphs but it does not explain well how the node types are handled differently from other methods. The objectives used to optimize the model do not seem to care about the node types.

**Reproducibility:**

4: Could mostly reproduce the results, but there may be some variation because of sample variance or minor variations in their interpretation of the protocol or method.

**Reviewer Confidence:**

4: Quite sure. I tried to check the important points carefully. It's unlikely, though conceivable, that I missed something that should affect my ratings.

---

> ### Author Rebuttal · Authors · 2023-08-29
>
> Thanks for the detailed comments. We have conducted experiments to illustrate the effectiveness and efficiency of the text augmentation strategy. We have also clarified the design on learning graph structure and heterogeneous node types and revised the typos in our paper. We have discussed the challenges of encoder-decoder and decoder-only PLMs in our work. We hope our answers can sufficiently address your concerns.
>
> **W1(part 1/2): Effectiveness of the text augmentation strategy.**
>
> We have conducted significant experiments and presented the p-values to support our claims. Specifically, we performed the node classification task multiple times (5 times) using different seeds ranging from 0 to 4 for these two pre-trained LMs whether using text augmentation strategy in our THLM.
>
> By comparing the results, we observed that all p-values were less than 0.05, indicating statistical significance and validating the accuracy gain achieved by our text augmentation strategy.
>
>
> | p-values  | MLP       | HetSANN   | RGCN      | RHGNN     |
> | --------- | --------- | --------- | --------- | --------- |
> | GoodReads | 1.432E-04 | 2.026E-04 | 1.440E-04 | 2.529E-04 |
> | Patents   | 1.940E-04 | 7.397E-04 | 3.678E-04 | 7.111E-04 |
>
> **W1(part 2/2): Efficiency of the text augmentation strategy.**
>
> We further report the training time and performance (Micro-Precision on node classification in the following table) of different methods on GoodReads and Patents. The running time of THLM on GoodReads (Patents) with aggregating neighbors ranging from 1 to 3 is 12,240s, 13,800s, 14,220s (12,611s, 14,472s, 15,078s). However, we also observe that THLM obtains the best performance with acceptable increments in computational complexity. Therefore, we conclude that THLM can achieve a good trade-off between efficiency and effectiveness.
>
> | Dataset   | Model           | MLP    | HetSANN | RGCN   | R-HGNN |
> | --------- | --------------- | ------ | ------- | ------ | ------ |
> | GoodReads | TAS(1-Neighbor) | 0.7480 | 0.8353  | 0.8421 | 0.8469 |
> |           | TAS(2-Neighbor) | 0.7547 | 0.8381  | **0.8426** | 0.8475 |
> |           | TAS(3-Neighbor) |**0.7549** | **0.8382**  | 0.8425 | **0.8485** |
> | Patents   | TAS(1-Neighbor) | 0.6959 | 0.7004  | 0.7211 | 0.7221 |
> |           | TAS(2-Neighbor) | **0.6960** | *0.7050*  | 0.7219 | 0.7233 |
> |           | TAS(3-Neighbor) | 0.6948 | **0.7050**  | **0.7275** | **0.7281** |
>
> Considering the statistical significance of the accuracy gain achieved through our text augmentation strategy and the careful consideration given to computational efficiency, we believe our approach offers a valuable trade-off between performance improvement and practical feasibility.
>
>
> **W2: Effectiveness of learning graph structure.**
>
> In our study, we focus on pretraining a language model that effectively incorporates both the heterogeneity and topology of graphs. To achieve this, we introduce a topology-aware pretraining task where the language model predicts nodes in the context graph surrounding a target node. This task optimizes both the language model and an auxiliary heterogeneous graph neural network (HGNN), allowing the model to utilize both first-order and high-order signals from the graph. To assess the ability of the HGNN to capture graph topology in our framework, we conduct an experiment where we replace the HGNN with a simple Multi-Layer Perceptron (MLP) mechanism and two state-of-the-art HGNNs (RGCN, R-HGNN) for pretraining and testing on downstream tasks.
>
> Compared to approaches such as LinkBERT and GIANT, which rely on self-supervised tasks to learn the graph structure, our method takes advantage of an auxiliary HGNN to incorporate the topology structure. By doing so, we are able to effectively capture the intricate topology inherent in graphs and enhance the pretraining process.
>
> The results presented in the following table, specifically the Micro-Precision on node classification, clearly demonstrate the significance of capturing graph topology. When our framework utilizes the HGNN to learn the inner structure, it achieves improved performance. On the other hand, using the MLP mechanism that neglects graph topology leads to a significant performance deterioration. Furthermore, by incorporating the auxiliary HGNN, our model surpasses the performance of both LinkBERT and GIANT.
>
> | Dataset   | Model     | MLP    | HetSANN | RGCN   | R-HGNN |
> | --------- | --------- | ------ | ------- | ------ | ------ |
> | GoodReads | w/ MLP    | 0.7528 | 0.8352  | 0.8376 | 0.8445 |
> |           | w/ RGCN   | 0.7608 | 0.8380  | 0.8411 | 0.8512 |
> |           | w/ R-HGNN | 0.7549 | 0.8382  | 0.8425 | 0.8485 |
> |           | LinkBERT  | 0.7244 | 0.8209  | 0.8259 | 0.8369 |
> |           | GIANT     | 0.6887 | 0.8098  | 0.8150 | 0.8258 |
> |           | THLM      | **0.7769** | **0.8439**  | **0.8437** | **0.8496** |
> | Patents   | w/ MLP    | 0.6903 | 0.6963  | 0.7201 | 0.7208 |
> |           | w/ RGCN   | 0.6911 | 0.6986  | 0.7184 | 0.7218 |
> |           | w/ R-HGNN | 0.6948 | 0.7050  | 0.7275 | 0.7281 |
> |           | LinkBERT  | 0.6333 | 0.6670  | 0.6968 | 0.6957 |
> |           | GIANT     | 0.6508 | 0.6701  | 0.6956 | 0.6999 |
> |           | THLM      | **0.7066** | **0.7159**  | **0.7324** | **0.7363** |
>
> **W3&Q1: Design for dealing with heterogeneous node types.**
>
> In our study, we address the heterogeneity of the graph, including different node types, by employing an auxiliary HGNN (Heterogeneous Graph Neural Network). Unlike existing approaches such as LinkBERT and GIANT, which focus on homogeneous graphs, we utilize an HGNN to effectively encode the heterogeneous graph. We have the flexibility to choose among various HGNNs, each with its own characteristics. For instance, RGCN (Relational Graph Convolutional Network) aggregates information across different types of nodes and edges, while R-HGNN (Relation-aware Heterogeneous Graph Neural Network) learns relation-aware node representations by integrating fine-grained representations within each relation and semantic information across different relations. The results obtained using "w/ RGCN" and "w/ R-HGNN" indicate that our model achieves better performance on downstream tasks when utilizing stronger HGNNs, as they provide more informative node representations.
>
> **Q2: Explaination of $\mathcal{A}$.**
>
> Thanks for pointing this out. $\mathcal{A}$ should be $\mathcal{U}$, which means that the number of node types and edge types should be more than 1 if the graph belongs to a heterogeneous graph.
>
> **Q3: Challenges for encoder-decoder and decoder-only PLMs.**
>
> We propose a new pretraining framework for LMs that incorporates topological and heterogeneous information from TAHGs to encode contextual representations. We focus on encoder-only models due to their proficiency in capturing context.
>
> Encoder-decoder PLMs, like T5, encode the input sentence and generate the output sequence using separate encoder and decoder components. However, in THLM, we only pretrain the encoder, which introduces inconsistency between the encoder and decoder, potentially affecting downstream tasks like machine translation. Decoder-only PLMs, such as GPT, generate the output sequence autoregressively during pretraining, but they may overlook essential information from the right context, leading to incomplete or less accurate semantic representations lacking global context consideration. Moreover, both PLM types prioritize sequence generation tasks, potentially introducing biases favoring fluent generation over accurate semantic representations.
>
> To address these challenges, techniques like incorporating bidirectional context, modifying training processes, or introducing additional objectives may be explored to improve the quality and accuracy of semantic representations generated by encoder-decoder and decoder-only models.

---

### Meta-Review · Area_Chair_VYjc · 2023-09-14

**Recommendation:** 4

**Metareview:**

**Strengths**:

1. The idea of pretraining using both text + graph info is interesting. Also, using graph info to handle text-richness imbalance is a good idea. It considers higher-order and heterogeneous structures.

2. The experiments were conducted extensively. New expts added as part of rebuttal process are relevant additions. Authors were also able to show that their method provides stat sig better results compared to methods already described in the paper and other recent methods like OAG-BERT.


**Weaknesses**:


1. Accuracy gains are not significantly large, considering the extra computations involved. But they are statistically significant.

2. Novelty is a little bit on lower side. As a reviewer indicates, earlier studies have tried combining language models and graph structures, and have handled imbalanced text in heterogeneous graph settings also.

3. Comparison with GraphFormers is missing.

**Suggestions**:


1. Please include a summary of several results presented in rebuttal, into the main paper.

2. Also, discussion on difference with Heterformer is nice. Please include it in related work.

3. Many clarifications during rebuttal are important for the reader. E.g., Clarification on the performance in Table 1, Table 2, and Table 3. Clarification on the performance between THLM and BERT. Please include them briefly in your main paper.

---

### Decision · Program_Chairs · 2023-10-07

**Decision:**

Accept-Findings

**Comment:**

**Strengths**:

1. The idea of pretraining using both text + graph info is interesting. Also, using graph info to handle text-richness imbalance is a good idea. It considers higher-order and heterogeneous structures.

2. The experiments were conducted extensively. New expts added as part of rebuttal process are relevant additions. Authors were also able to show that their method provides stat sig better results compared to methods already described in the paper and other recent methods like OAG-BERT.


**Weaknesses**:


1. Accuracy gains are not significantly large, considering the extra computations involved. But they are statistically significant.

2. Novelty is a little bit on lower side. As a reviewer indicates, earlier studies have tried combining language models and graph structures, and have handled imbalanced text in heterogeneous graph settings also.

3. Comparison with GraphFormers is missing.

**Suggestions**:


1. Please include a summary of several results presented in rebuttal, into the main paper.

2. Also, discussion on difference with Heterformer is nice. Please include it in related work.

3. Many clarifications during rebuttal are important for the reader. E.g., Clarification on the performance in Table 1, Table 2, and Table 3. Clarification on the performance between THLM and BERT. Please include them briefly in your main paper.